# Is the Invasiveness of *Pittosporum undulatum* in Eucalypt Forests Explained by the Wide Ranging Effects of Its Secondary Metabolites?

Dalila Pasquini [1,2], Luana Beatriz dos Santos Nascimento [1], Cecilia Brunetti [2,*], Francesco Ferrini [1,3] and Roslyn M. Gleadow [4]

1    Department of Agriculture, Food, Environment and Forestry (DAGRI), University of Florence, 50144 Florence, Italy
2    National Research Council of Italy, Institute for Sustainable Plant Protection (IPSP), 50019 Sesto Fiorentino, Italy
3    VALUE Laboratory on Green, Health & Wellbeing, University of Florence, 50019 Sesto Fiorentino, Italy
4    School of Biological Science, Monash University, Melbourne, VIC 3800, Australia
*    Correspondence: cecilia.brunetti@ipsp.cnr.it

**Abstract:** Climate change is a driver of biodiversity loss, often favouring invasive species such as in the case of *Pittosporum undulatum* in *Eucalyptus* forests of south-eastern Australia. We tested whether the invasiveness of *P. undulatum* is due to the release of secondary metabolites with allelopathic action inhibiting other species germination or to the content of antioxidant secondary metabolites. We compared the germination of *P. undulatum* and *Eucalyptus ovata* seeds on different substrates watered with different leachates. Polyphenolic compounds, saponins and tannins of *Eucalyptus* spp. and *P. undulatum* leaf extracts were quantified. Biogenic Volatile Organic Compounds were collected in the field and analyzed to compare the emissions in eucalypt forests with and without *P. undulatum*. *Eucalyptus* germination rates were not affected by different leachates and no allelopathic compounds were identified in *P. undulatum* leachate. Flavonoids and tannins characterized *Eucalyptus* leachates, while *P. undulatum* leachates showed high hydroxycinnamic acids content. The forests invaded by *P. undulatum* were characterized by high levels of monoterpenes, whereas the forest lacking *P. undulatum* were dominated by sesquiterpenes. Our results suggest that the invasiveness of *P. undulatum* may be due to the high content in secondary metabolites that play a protective role against abiotic stresses rather than the release of allelopathic compounds.

**Keywords:** allelopathy; biodiversity; climate change; environmental stresses; germination test; secondary metabolites

## 1. Introduction

Changing climatic conditions have played a major role in the evolution of plants, which may develop defense strategies to help them survive new suites of stressors [1]. Across Mediterranean climate regions around the world (e.g., Mediterranean basin, Southwest and South Australia, the Cape Region in South Africa, California in the USA, and Central Chile) [2], the general increase in the temperature and reduction in precipitation, associated with high human-driven impacts on land, is leading to a wide change in the richness and distribution of many species [3,4]. Indeed, these regions are experiencing a change at the environmental level influenced by abiotic and biotic stresses, resulting in consequences in biodiversity which may impair ecosystem function [5]. Forests and land ecosystems have been subject to nutrient losses and increases in the frequency and severity of extreme climatic events [6]. Furthermore, an increase in $CO_2$ concentrations, acid deposition on soil as well as diseases and pathogens have also occurred [1]. Such changes may lead to an imbalance between the mortality of native plants and invasion of exotic species,

the latter sometimes being more competitive in the new and/or degraded habitats [7]. Species invasion (either by indigenous or exotic species) is a serious threat to natural environments [8], posing difficult challenges to ecological management worldwide [9].

Invasive species can modify the composition, structure and functionality of native plant communities [10] and can also influence the ecosystem with the production and release of allelopathic compounds [11]. Despite several studies stating that disturbance of natural ecosystems is a precursor of exotic species invasion [12,13], other studies conducted in south-eastern Australia have demonstrated that native trees and shrubs can also promote decline in species richness, since they may possess invasive characteristics [14,15]. One of these native species, which is often considered to be an aggressive invader in several areas of south-eastern Australia, is *Pittosporum undulatum* Vent. [14,16,17].

*Pittosporum undulatum* (Sweet Pittosporum) is an evergreen tree (~5–15 m tall) native to south-east Australia, with a natural distribution spanning from south-east Queensland to eastern Victoria that has a high invading potential thanks to its strong capacity to colonise different habitats [18–20]. In fact, *P. undulatum* is a notorious invader of forests around the world, and it is also spreading outside its natural range in Australia [14,21]. Indeed, in addition to the already known invasions in Hawaii, Bermuda, Canary Islands, New Zealand, Lord Howe Island, Norfolk Islands, and other parts of Australia, some authors have reported the spread of this species through US, Mexico, Guatemala, the Caribbean (Jamaica and Puerto Rico), South America (Colombia, Ecuador, Bolivia and Brazil), South Africa, Spain, and Portugal [22]. Wet and dry sclerophyll forests, dominated by *Eucalyptus*, are invaded by *P. undulatum* right across the continent and causing a serious reduction in floristic and structural diversity [23] and bird assemblages [24]. This phenomenon is threatening the survival of the natural stands of mixed eucalypt woodlands [14] characterized by *Eucalyptus polyanthemos* (Red Box), *Eucalyptus goniocalyx* (Long-leaf Box), *Eucalyptus ovata* (Swamp Gum), and *Eucalyptus rubida* (Candlebark), evergreen small/medium-size tree species all native to south-eastern Australia (southern New South Wales to Victoria) [25].

The invasive capacity of *P. undulatum* is linked to a wide range of favourable traits, such as high germination capacity, high competitiveness of its seedlings, and a dense crown with dark evergreen leaves blocking up to 90% and 75% of sunlight during winter and summer, respectively [14,20,21,23,26]. In autumn, it produces orange capsules, usually carrying around 20 sticky seeds [27], which allow its diffusion capacity by animals. The main characteristics of *P. undulatum* that could explain its spread are its high germination rates [28] and its capacity to grow in environments altered by human activity and without the need of forest fires [22]. Additionally, it has a great adaptability to a wide range of climatic conditions, and a root system capable of growing in different edaphic conditions [14,28]. Instead, the regeneration cycle of many eucalypt species is linked to fire, a natural factor in the Australian environment. Fire releases the seed from the canopy (there is little seed stored in the soil) and increases the amount of light reaching the forest floor. The absence of fires in peri-urban areas, due to fire-suppression policies carried out since 1939 [14,18,28], might therefore have encouraged *P. undulatum*, which is very drought tolerant at the seedling stage [26], at the expense of *Eucalyptus* species. In addition, *Eucalyptus* species included this study are all known to be able to resprout from epicormic buds after even very hot fires, whereas fires eliminate seedlings and the local seed bank of *P. undulatum* [18].

The invasive capacity of *P. undulatum* has also been linked to chemical traits. For example, aqueous extracts of *P. undulatum* leaves have been demonstrated to have enough allelopathic action against roots and seedlings of other species for the authors to hypothesize their possible use as a natural herbicide [29]. The highly competitive nature of *P. undulatum* has also been attributed to the inhibition of germination and growth of other species due to the presence of allelopathic compounds belonging to the terpenoids, alkaloids, glycosides, flavonoids, saponins, and tannins classes that are stored in the leaves and emitted as volatiles [30,31]. It is well known that *P. undulatum* leaves contain high concentrations of saponins [14,32]. Nevertheless, a study of secondary metabolites stored and emitted against

autochthonous species in its native habitat has yet to be conducted. Since Gleadow and Rowan [21] reported that the success of *P. undulatum* spread is primarily due to the survival of the seedlings and not to their growth rate, we also hypothesized a role of the different classes of secondary metabolites—both compounds stored in the leaves and volatile organic compounds released in the atmosphere—on its capability to respond to abiotic stresses and survive under stressful conditions exacerbated by climate change in addition to their purported role in allelopathy.

This study aims to elucidate possible biochemical bases of *P. undulatum* invasiveness in a *Eucalyptus* sclerophyll forest located in south-eastern Australia. Firstly, we tested the hypothesis that the success of *P. undulatum* is partly due to a reduction of seed germination capacity of native plants, such as the co-occurring tree, *Eucalyptus ovata*. To accomplish this, we compared the germinability of *P. undulatum* and *E. ovata* on different substrates watered with various leaf/litter leachates. We then tested whether differences in secondary metabolites, both stored and emitted, between *P. undulatum* and *Eucalyptus* spp. helped explain the high degree of invasiveness of *P. undulatum*. We also evaluated whether the accumulation of carbon-based secondary metabolites in leaves may help plants to thrive under multiple environmental stresses [33]. In detail, we carried out: (i) germination analyses to test the allelopathic action of different leachates obtained from litter and green leaves of *P. undulatum*, and from litter of *Eucalyptus* spp.; (ii) analyses of secondary metabolites (saponins, tannins and polyphenols) in leaf and litter extracts of *P. undulatum* and *Eucalyptus* spp.; and finally, (iii) analyses of Biogenic Volatile Organic Compounds (BVOCs) at the environmental level, using Solid Phase Microextraction (SPME) fibres to observe possible differences in volatile profiles between the two areas in pure *Eucalyptus* spp. woodlands and in woodlands invaded by *P. undulatum*.

## 2. Materials and Methods

### 2.1. Study Area

The study site is the Bunjil Reserve (north-east of Melbourne, Victoria, Australia, Figure 1A–C), one of the seven Panton Hill Bushland Reserves. The seven reserves cover an area of 140 hectares, extending from Smiths Gully, in the north, to Watsons Creek, in the south. The annual average precipitation for the area is 660 mm, with the most rain falling in November (73.3 mm), and the driest month being March (42.7 mm). Mean warmest temperatures are approximately 27.1 °C and 14.5 °C for summer and winter, respectively (according to meteorological data acquired by the Australian Bureau of Meteorology Station 086068, situated in Viewbank, approximately 15 km away from the Bunjil Reserve). The reserve contains two different conditions: an area invaded by *P. undulatum* (I), nearby Bishops Rd (37°38′31.94″ S, 145°14′36.14″ E), around 700 m from the Bunjil Reserve gate; and an area with high-quality remnant native vegetation, (R) (37°38′49″ S, 145°14′54.75″ E). Due to their close proximity (approximately 800 m apart) the two areas present the same climatic conditions, and same altitude: the first area is at 174 m a.s.l., while the second is at 153 m a.s.l. The invaded area at Panton Hill was characterised by the presence of *P. undulatum* (ca. 40% of woody species), *Eucalyptus* spp. (*E. goniocalyx, E. polyanthemos, E. ovata*) (ca. 55% of woody species) and *Acacia* spp. (ca. 5% of woody species) (Figure 1D). The adult *P. undulatum* trees were healthy, with lower crowns than those of the nearby eucalypts and acacias present, but with a greater density (almost forming an impenetrable green wall). *Eucalyptus* and *Acacia* trees were also healthy, and the soil was flat and rich in grasses. This vegetation composition resulted in a more humid microclimate than the one commonly found in sclerophyll forests, as reported in other works carried out in Victoria [14]. The natural area (R) was characterised by extensive stands of mixed eucalypt woodland (95%–98% of *Eucalyptus* spp.: *E. rubida, E. goniocalyx, E. polyanthemos, E. ovata*) and 2%–5% of *Acacia* spp. (Figure 1E). No presence of *P. undulatum* was found in the second area (R) and the soil was slightly sloping, almost bare and drier, with little presence of herbaceous plants. It can be assumed that the dominant factor between these two areas is the presence–absence of *P. undulatum*. *Eucalyptus* species are noteworthy in maintaining high genetic diversity even

in fragmented woodland populations [34]. Moreover, the trees in this study pre-date the fragmentation of the landscape and the disruption by invading *P. undulatum*. There has never been a record of a *Eucalyptus* seedling growing under a canopy of *P. undulatum* in the field in the forty years that this has been the subject of study [10,14,18].

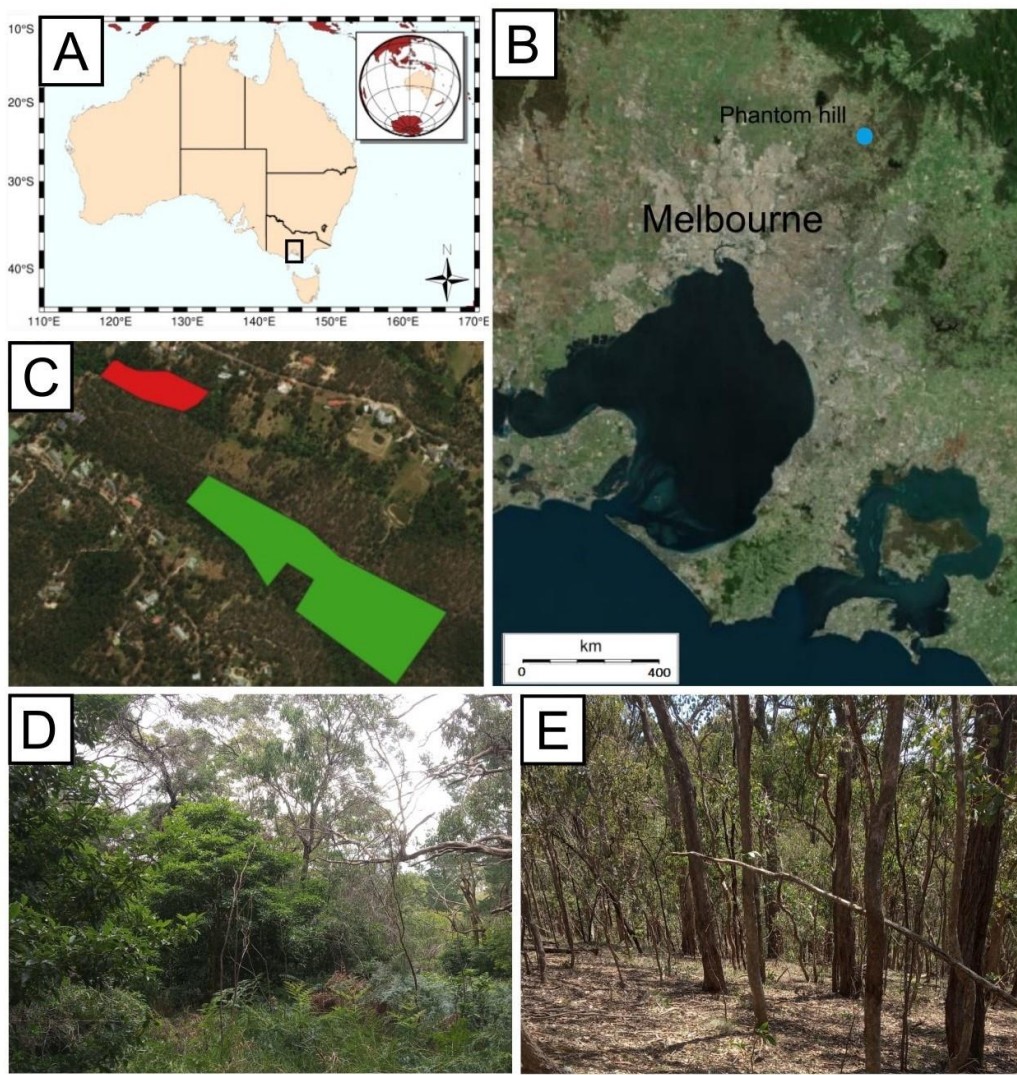

**Figure 1.** Location of the study sites. (**A**) location of Greater Melbourne Area within Australia. (**B**) location of the study area of Phantom Hill (light-blue dot) in respect to the city of Melbourne. In the panel (**C**), the red area represents the study site with the presence of *P. undulatum* (denominated I, invaded), and the green area represents the site characterised by the presence of *Eucalyptus* spp. and *Acacia* spp. (denominated R, remnant vegetation). A picture of the areas I (**D**) and R (**E**).

*2.2. Germination Experiment*

A germination study was conducted to test whether leaf allelopathic chemicals influence germination success or germination trajectories, as proposed by Gleadow [23]. To accomplish this, natural soil was collected in situ from the invaded and reference (*Eucalyptus*-dominated) areas in the study site at Phantom Hill (Figure 1). This soil was used as the substrate base in Petri dishes to create nature-like germination conditions and to understand whether there could be possible allelopathic compounds released in the soils that could interfere with germination rates [35]. Thus, we have not sterilised the soil, to avoid altering it in any way. Seeds, without pre-germinative treatments, were directly placed on filter papers laid over the soil and watered with leachates of green leaves and litter from *P. undulatum* and *Eucalyptus* spp., and compared to seeds watered only with

water (details below). The field sampling was conducted on the 20 November 2019, during the Austral spring season. Seeds were sown on 25 November 2019.

In detail, within the invaded area we collected: a total of 500 g of soil (layer 15–30 cm deep) pooled together from 10 sampling points underneath *P. undulatum* plants, approximately 250 g of *P. undulatum* green leaves and 250 g of *P. undulatum* litter. Likewise in the reference area we collected: a total of 500 g of soil (layer 15–30 cm deep) pooled together from 10 sampling points underneath under the mixed *Eucalyptus* canopy and approximately 250 g of *Eucalyptus* litter. For the latter, only leaf litter was collected, as it has been demonstrated that the litter of eucalypt has high concentration of allelopathic compounds compared with other plant parts [36]. All leaves (green and litter) were placed in plastic bags and stored at −20 °C overnight. The following day, soil samples were sieved using a 2 mm sieve and the leachates were prepared as follows: 50 g of green *P. undulatum* leaves, 50 g of *P. undulatum* litter and 50 g of *Eucalyptus* spp. litter were washed twice with distilled water, put separately in 2500 mL volumetric flasks without tissue disruption, and filled up to 1000 mL with distilled water. After 72 h of soaking at room temperature (21–22 °C), the 5% aqueous extracts were filtered and stored at 4 °C until use [37].

A pool of *P. undulatum* seeds was collected from the ground under several naturally sown trees growing on the Clayton campus of Monash University (Melbourne, Victoria). Seeds of *E. ovata* were obtained from the Australian Tree Seed Centre (CSIRO, ACT, Australia, Seedlot 20808), *E. ovata* was selected because its distribution overlaps with both the native and *P. undulatum* range and seeds were readily available.

Germination tests were carried out using three substrates and four watering treatments in a factorial design. The three substrates were: filter paper alone (control); 5 g of soil collected under *Eucalyptus* trees; and 5 g of soil collected under *P. undulatum* plants. The four watering treatments were: distilled water (control); leachate of *P. undulatum* green leaves ($L_{PG}$); leachate of *P. undulatum* litter ($L_{PL}$); and leachate of *Eucalyptus*. spp. litter ($L_{EL}$). Thus, the experimental design consisted of 12 combinations (3 substrates × 4 watering treatments) per species and three replicates for each combination, giving a total of 72 Petri dishes (90 mm, Filter paper Advantech type 2) and 2160 seeds (30 seeds per dish). Each replicate consisted of a Petri dish. Dishes were placed in an incubator with at a constant temperature of 20 °C [38], a light intensity of 200 µM quanta m$^{-2}$ s$^{-1}$ and a photoperiod of 12 h light/12 h darkness [39]. All seeds were sowed on the 25th of November. Daily visual inspections were carried out to count germinated seeds. Seeds were considered to have germinated as soon as the embryo ruptured the seed coat [40] and the radicle was visible. Additionally, the state of cotyledons and seedlings tissues were noted. At the end of the experiment (22 days for *E. ovata* and 38 days for *P. undulatum*) the seeds were removed from the Petri dishes; the length of hypocotyl, root and cotyledons for each seedling was also noted; and a squash test was conducted at the end of the experiment to detect the vitality of the seeds that did not germinate [40].

Using the function *SSlogis()* in the '*nlme*' R package [41], for each species and combination of substrate and watering treatments, three parameters were estimated:

- Germination Percentage (*GP*), a numeric parameter representing the top horizontal asymptote of the S-shape curve, signifying the total germination obtained, where 0 corresponds to no germination and 1 is the full germination (*Asym* in R);
- $t_{50}$, a numeric parameter representing the day of the inflection point of the curve, i.e., where the germination is half of the total final germination (*Xmid* in R);
- $t_{75}$, a numeric scale parameter obtained from the number of days between 3/4 of *GP* and $t_{50}$, representing the growth rate during the exponential phase (*Scal* in R).

The three parameters (i.e., *GP*, $t_{50}$ and $t_{75}$) were used to model the S-shape germination curves for each species and combination of substrate and watering treatments according to the equation presented by Pinheiro and Bates [41]:

$$y = \frac{GP}{1 + exp[-(days - t_{50})/t_{75}]} \tag{1}$$

where *days* are the time of observations.

In addition, using the package "*germinationmetris*" [42], the following indices were calculated:

- the time for the first germination ($t_0$) [43];
- the time for the last germination ($t_f$) [43];
- the time spread of germination, was calculated as the difference between the last and first day of germinations ($t_f - t_0$) [43];
- the Vigor Index (*VI*) of seedlings was measured as

$$V I = GP \times (L_r + L_s) \tag{2}$$

where *GP* is the germination percentage and $L_r$, $L_s$ are root and shoot lengths of seedling, respectively [44].

### 2.3. Analyses of Total Saponins and Total Condensed Tannins in Leaf Litter Leachates

Total saponins and total condensed tannins contents were measured on $L_{EL}$ and $L_{PL}$, on ethanolic extracts of *Eucalyptus* spp. green leaves and litter (i.e., $E_{EG}$, $E_{EL}$, respectively) and on ethanolic extracts of *P. undulatum* green leaves and litter ($E_{PG}$, $E_{PL}$, respectively), prepared as reported in the following Section 2.4. The green leaves and the litter were collected the same day during the field sampling conducted on the 20 of November 2019 and utilized to obtain the leachates used for the germination test. For the leachates, before performing the spectrophotometric assays, 3 mL of each sample was evaporated under vacuum and re-dissolved in 3 mL of ethanol. All the analyses, both for saponins and tannins, were conducted in triplicate.

The total saponin content (TSC) was measured following the procedure described by Le et al. [45]. In detail, 0.15 mL of sample extract, 0.15 mL of vanillin in ethanol (8% *w/v*) and 1.5 mL of sulphuric acid in water (72% *v/v*) were mixed and placed in a warm bath at 60 °C for 15 min. The TSC was determined using UV/VIS spectrophotometer (Lambda 25, Perkinelmer) at 535 nm. Diosgenin (Extrasynthese, Genay Cedex France) was used as an external standard to create a five-points calibration curve (0.025–0.25 mg/mL) and the TSC obtained was expressed as milligram diosgenin equivalents (mg DE) per g of Dry Weight (DW) of plant material.

The total condensed tannin content (TcTC) was measured following the protocol described by St-Pierre et al. [46], adding 1 mL of a solution composed of 0.1% of 4-dimethylaminocinamaldehyde (DMCA) in methanol-HCl 9:1 (*v/v*) to each extract (water-based and ethanol). The mixture was mixed vigorously for 1 min and then incubated in the dark at room temperature, for 15 min. The TcTC was determined using UV/VIS spectrophotometer at 640 nm. Epicatechin (Extrasynthese, Genay Cedex France) was used as external standard to create a seven-point calibration curve (0.0025–0.1 mg/mL) and the TcTC was expressed in catechin equivalents (mg CE) per g of DW.

### 2.4. HPLC Analysis of Polyphenols in Leaves and Leaf Litter

Lyophilized powdered samples (green leaves and litter collected during the field sampling conducted on the 20 November 2019) of *P. undulatum* and *Eucalyptus* spp. were weighed (150 mg), placed into test tubes, and added with 5 mL of a ethanol:water solution (80:20, *v/v*), acidified to pH 2.5 with 0.1% of HCOOH. Each sample was sonicated in an ultrasonic bath for 20 min, and the entire procedure was replicated three times. After the extraction, the supernatant was defatted four times by adding 3 mL of n-hexane. Of the 15 mL of ethanolic extract, 3 mL were used for the spectrophotometer analyses of saponins and tannins (mentioned above, Section 2.3) and the remaining 12 mL were evaporated under vacuum and re-dissolved in 1 mL of methanol:water (50:50, *v/v*). An aliquot of 5 μL of the extracts was injected into the Perkin® Elmer Flexar liquid chromatograph equipped with a quaternary 200Q/410 pump and coupled with a LC 200 diode array detector (DAD) (Perkin Elmer®, Bradford®, CT, USA). The separation was achieved on a Zorbax® SB-18

column (250 × 4.6 mm, 5 μm) (Agilent, Santa Clara, CA, USA), kept at 30 °C. The mobile phase consisted of acidified water (pH 2.5 adjusted with HCOOH; solvent A) and acidified acetonitrile (pH 2.5 adjusted with HCOOH; solvent B). The gradient used was similar for all extracts: 97% of solvent A and 3% of solvent B (0–10 min); minutes 10–11 of hold time; 60% of solvent A and 40% of solvent B (12–66 min); minutes 67–71 of hold time; and 97% of solvent A and 3% of solvent B (71–72 min). The flow rate was kept constant at 0.6 mL min$^{-1}$. All the analyses were performed in triplicate, recording the spectra from 180 to 900 nm, setting the wavelengths used to quantify the different compounds at 280 nm and 350 nm for *Eucalyptus* spp. extracts, and 280 nm and 330 nm for *P. undulatum.*

The identification of each compound was based on a combination of retention time and spectral matching, with data comparison against authentic standards (gallic, caffeic and p-coumaric acids, all from Sigma-Aldrich Chemie, Darmstadt, Germany; and ellagic acid, rutin, luteolin-7-O-glucoside, all from Extrasynthese, Genay Cedex France) and literature. The quantitative results of polyphenols were reported as mg/g of dry weight (DW) and expressed as the sum of the content of individual compounds belonging to each phenolic class: Total Gallo Ellagic Tannins Content (TTC), Total Flavonoid Content (TFC) and the Total Hydrocinnamic Acid derivatives Content (THC).

### 2.5. Collection and Analysis of BVOCs

During the same sampling day for the collection of materials for the germination experiment (20 November 2019), five Solid Phase Microextraction (SPME) fibres were used at each sampling site to collect Biogenic Volatile Organic Compounds (BVOCs) at environmental level. The local temperature was 37 °C, with 29% humidity, and 13 km/h wind speed with a NE direction. Ten fan-samplers mounted with SPME fibres (Sigma-Aldrich of 2 cm and assembly Divinylbenzene/Carboxen /Polydimethylsiloxane) were installed at a height of 45 cm from the ground [47]. This height was chosen to simplify the sampling, since terpene concentrations have been shown to be higher at heights from 0 to 4 m [48]. Sampling time was set between 11 a.m. and 3 p.m. The fibres were then put in a special tray, within a hermetic case and dedicated Teflon pressure supports to seal the needles for the transport to the laboratory.

The SPME fibres were desorbed in an Agilent 7890 B gas chromatograph coupled with a 5977A mass spectrometer with EI ionization operating at 70 eV. A chromatographic column Agilent DB-Wax 60 m × 250 μm × 0.5 μm was used. The injector temperature was set to 260 °C, splitless mode, with a flow of 1.2 mL/min. The oven temperature program consisted of an initial temperature of 40 °C for one minute, which was then increased by 5 °C/min until 210 °C, and then by 10 °C/min until 250 °C (max temperature for this column). Lastly, the temperature was decreased to 240 °C and was held for 10 min, resulting in a total run time of 51 min. The lowest mass acquired was 29 *m/z* and the highest was 350 *m/z* at three scans per second.

The data was analysed using the Agilent Mass Hunter software (Qualitative Analysis-Version B.06.00; Quantitative Analysis Version B.07.01/Build 7.1.524.0), and the terpenes were putatively identified by matching their mass spectra and retention indices with those reported in the NIST 11 spectral database library. Information related to the fragmentation patterns and retention times available from scientific literature and authentic standards was used for the final compound annotation [49]. The amount of monoterpenes and sesquiterpenes, expressed as peak areas, were related to Total Ion Current (TIC).

### 2.6. Statistical Analyses

All statistical analyses were carried out using R (version 4.1.0) and RStudio (version 1.4.1717) and the analytical process was as follows.

i.    A Nonlinear Mixed-Effects Models analysis was used on each daily cumulate count (for each combination of substrate and watering treatments) given their non-linear trends over time. The starting estimates of the S-shape curves were estimated through the *SSlogis()* function (*nlme* library) and the model space was investigated by com-

paring marginal models [41] to select the most parsimonious model. Two non-linear models were fitted on the data: one model for *Eucalyptus ovata* seeds and another for *Pittosporum undulatum* seeds.

The fixed part of the most parsimonious models was in the form

$$GP + t_{50} + t_{75} \sim \text{substrate} \tag{3}$$

and

$$GP + t_{50} + t_{75} \sim \text{substrate} \times \text{treatment} \tag{4}$$

for *Eucalyptus ovata* seeds and *Pittosporum undulatum*, respectively.

The random part was, for both models, in the form

$$GP + t_{50} + t_{75} \mid \text{Dishes} \tag{5}$$

where $GP$, $t_{50}$ and $t_{75}$ are the parameters described above (see Section 2.2) and Dishes represents the number of the dishes ($n = 36$). Thus, in the fixed part, the three parameters are influenced by the substrate in the case of *Eucalyptus ovata*. The watering treatments and the relative combinations with substrates were not considered, since the exploration of marginal models showed no treatment significance. In the case of *P. undulatum* seeds the three parameters are influenced by the interaction of substrate and treatment.

ii. Each index (i.e., $t_0$, $t_f$, $t_f - t_0$) was calculated at the end of the germination experiment for every single species. Counts data were fitted using a General Linear Model (*glm()*, requiring the *lattice* and *faraway* packages [50]) using a Poisson distribution family with log-link function (*glm (INDEX ~ Substrate × Treatment, family = poisson (link = "log")))*), and the significance was calculated on the exponents and not on the values of the indices. The effect of different substrates (Substrate) and leachates (Treatment), their interaction on germination, and seedling development was observed. The models were carried out with the Petri dishes characterized by filter paper and distilled water as reference (Intercept). Finally, the models were chosen after checking for overdispersion.

iii. A two-way Analysis of Variance (ANOVA) was conducted for the continuous data obtained from the Vigor Index (VI), observing the interaction *substrate × treatment*. Before carrying out the ANOVA, the assumption of normality and homoscedasticity were checked using Shapiro and Levene's tests, respectively [51,52]. Finally, a Tukey post-hoc test was conducted.

iv. For continuous variables (TSC, TcTC, TTC, TFC and THC), a one-way non-parametric analysis of variance (Kruskal–Wallis Test) was conducted. This test was carried out, since the ANOVA's assumptions of normality tested with Shapiro's were not met, while the heteroscedasticity tested with Levene's test was met. After that, Dunn's Multiple Comparison post-hoc test was carried out.

v. For BVOCs compounds, in order to test differences between the two studied areas (I and R), we calculated the relative amount of each monoterpene (MT) and sesquiterpene (SQT) identified, expressed as a percentage of total terpenes peak areas obtained by GC-MS (TMTs + TSQTs) for both areas. The mean percentages of each terpene were analysed by a one-way analysis of variance.

All results were considered significant when $p < 0.05$.

## 3. Results

### 3.1. Leachates and Substrates Effects on the Germination of P. undulatum or E. ovata

The germination curves showed that the seeds were able to germinate under all the combination of substrates and treatments, with no signs of total inhibition, but with different percentages and velocity for each species (Figures 2 and 3).

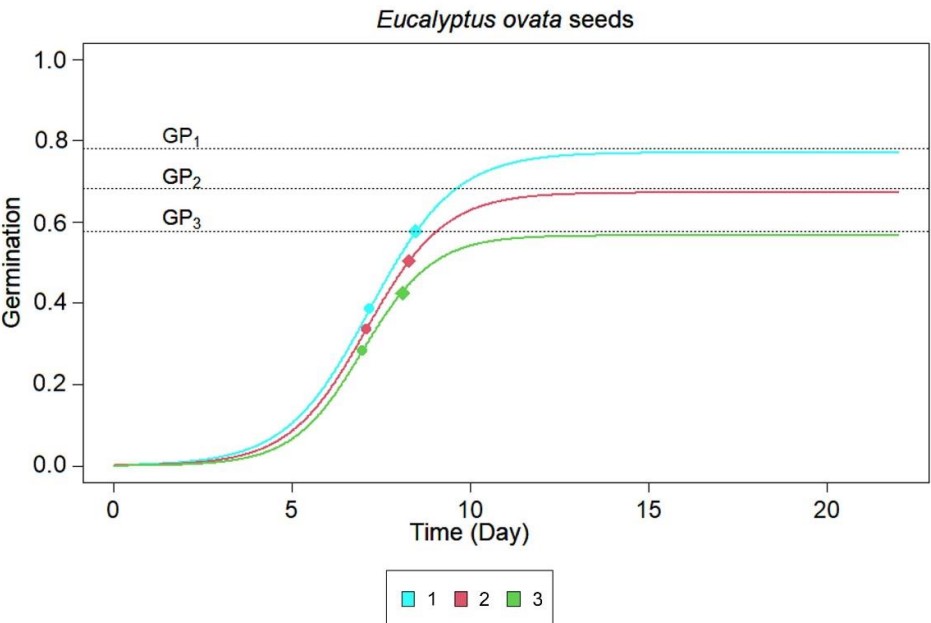

**Figure 2.** Temporal progression of *Eucalyptus ovata* seeds germination (range 0–1) for three substrates: 1—Filter paper; 2—*P. undulatum* soil; 3—*Eucalyptus* soil. On the *x* axis the days of the experiment (22 days) are reported. The horizontal asymptotes represent the total final germination ($GP_i$) carried out for each substrate (i = 1–3), the coloured dots represent the inflection point (i.e., $t_{50}$) for each curve and the coloured diamonds represent $t_{75}$ (i.e., the time required to reach 3/4 of $GP$ from relative $t_{50}$).

In the case of *Eucalyptus ovata* seeds, the starting observations and estimates of the S-shape curves ($n = 36$, Figure S1) were observed, and the comparison of marginal models reported that there was no significant difference in the watering treatments and the relative combinations of substrates. For this reason, only three S-shape curves are presented in Figure 2. When looking at Figure 2, it is possible to observe the temporal progression of the germination for *Eucalyptus ovata* seeds in the three substrates (i.e., filter paper, *P. undulatum* soil and *Eucalyptus* soil) and the three curves have three different $GP$ (asymptotes—horizontal lines) and three $t_{50}$ (inflection points—coloured dots).

The Petri dishes with higher values of $GP$ and $t_{50}$ were the dishes with filter paper as substrate, while the lower values of $GP$ and $t_{50}$ belong to those with *Eucalyptus* soil substrate. In Table 1, the explicit values of the three parameters, obtained from the nlme model using *SSlogis()*, are reported, with their level of significancy in superscript. The nlme analysis showed that both the S-shape curves, obtained for the Petri dishes with soil substrates, were significantly different ($p < 0.05$) with respect to the Petri dishes with filter paper. The only exception was the $t_{75}$ parameter for the Petri dishes with *Eucalyptus* soil (representing the growth rate during the exponential phase of the curve) that showed no significant differences (ns).

**Table 1.** Germination of *Eucalyptus ovata* seeds under different substrates (i.e., filter paper; *P. undulatum* soil; *Eucalyptus* soil). In the $GP$, $t_{50}$ and $t_{75}$ columns are reported the values obtained from the Nonlinear Mixed-Effects Model '($GP + t_{50} + t_{75}$ ~ substrate)' with the function SSLogis() and in superscript are noted the significance levels obtained from nlme() (ns > 0.05, * 0.01 < $p$ < 0.05, *** $p < 0.001$) (the relative summary are reported in Table S1).

| Substrate | $GP$ (%) | $t_{50}$ (Day) | $t_{75}$ (Day) |
|---|---|---|---|
| Filter paper (Intercept) | 0.77 | 7.18 | 1.19 |
| *P. undulatum* soil | 0.67 * | 6.26 *** | 0.80 *** |
| *Eucalyptus* soil | 0.57 *** | 6.19 *** | 1.04 [ns] |

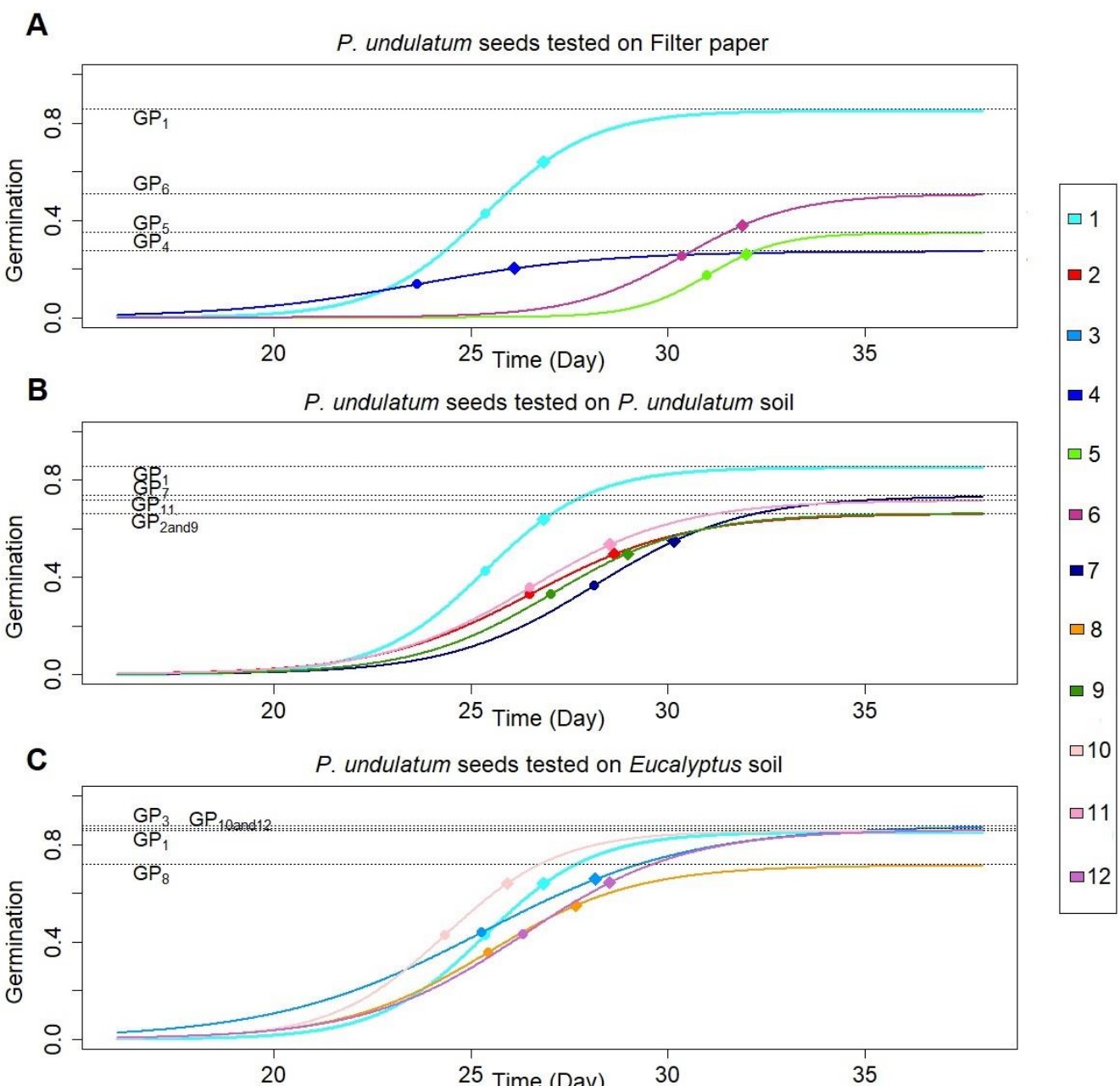

**Figure 3.** Panel (**A**) shows the overall temporal progression of *Pittosporum undulatum* seeds germination for all combinations on Filter paper substrate: 1—Filter paper Water; 4—Filter paper $L_{EL}$; 5—Filter paper $L_{PG}$; 6—Filter paper $L_{PL}$. Panel (**B**) shows the curves obtained from seeds germinated on *P. undulatum* soil substrate: 2—*P. undulatum* soil Water; 7—*P. undulatum* soil $L_{EL}$; 9—*P. undulatum* soil $L_{PG}$; 11—*P. undulatum* soil $L_{PL}$ and the reference curve (i.e., 1). Panel (**C**) shows those obtained from seeds germinated on *Eucalyptus* soil substrate: 3—*Eucalyptus* soil Water; 8—*Eucalyptus* soil $L_{EL}$; 10—*Eucalyptus* soil $L_{PG}$; 12—*Eucalyptus* soil $L_{PL}$ and the reference curve (i.e., 1). On the *x* axis are reported the days of the experiment (from 15 to 38 day—the first two weeks were removed since the first germination occurred at day 15). The horizontal asymptotes represent the total final germination ($GP_i$) carried for each S-shape curves (i = 1–12), the coloured dots represent the inflection point (i.e., $t_{50}$) for each curve and the coloured diamonds represent the time to arrive at 3/4 of *GP*. The third parameter, $t_{75}$, is calculated by the time between $t_{50}$ and 3/4 of *GP*.

In the case of *Pittosporum undulatum* seeds, the initial observations of all the S-shape curves ($n$ = 36; Figure S2) and the comparison of marginal nlme models showed a significant interaction between substrates and watering treatments. In Figure 3 it is possible to observe the 12 germination curves obtained for each combination of substrate $\times$ treatment. The curves 4, 5, and 6 (i.e., 4—Filter paper $L_{EL}$; 5—Filter paper $L_{PG}$; 6—Filter paper $L_{PL}$) were those more different from the reference curve (i.e., 1—Filter paper Water). These three had the lowest *GP* parameter and the curves 4 and 5 had very high $t_{50}$ values. For better visualisation, Figure 3 has been split in three different panels (i.e., A, B, C). In each of the three panels, the germination curve modeled with nlme from the dishes with filter paper substrate and distilled water as watering treatment was kept as reference, and then the other S-shape curves were grouped according to the different substrates (i.e., panel A—Filter paper; panel B—*P. undulatum* soil; panel C—*Eucalyptus* soil). In Table 2 are reported the values of the three parameters (i.e., *GP*, $t_{50}$ and $t_{75}$) and their significance levels are shown in superscript. In Table 2, only the Petri dishes characterized by *Eucalyptus* soil, as substrate, and distilled water, as watering treatment, did not show a *GP* significantly different from the Intercept (i.e., filter paper and water), while all the others were significantly different. Observing the second parameters (i.e., $t_{50}$), only the Petri dishes with distilled water as watering treatments presented no significant differences (*ns*) with respect to the Intercept. Finally, for the last parameter (i.e., $t_{75}$), the three Petri dishes characterised by $L_{PL}$ watering treatments and the one with *P. undulatum* soil as substrate and $L_{PG}$ as watering treatment resulted in no statistical significances, with *p*-values higher than 0.05.

**Table 2.** Germination of *Pittosporum undulatum* seeds under different substrates (i.e., Filter paper; *P. undulatum* soil; *Eucalyptus* soil) and different watering treatments (i.e., Water; $L_{EL}$ = Leachate of *Eucalyptus* spp. Litter; $L_{PG}$ = Leachate of *P. undulatum* Green leaves; $L_{PL}$ = Leachate of *P. undulatum* Litter). In the *GP*, $t_{50}$ and $t_{75}$ columns are reported the values obtained with the Nonlinear Mixed-Effects Model '(*GP* + $t_{50}$ + $t_{75}$ ~ substrate $\times$ treatment)' with the function SSLogis() and in superscript are noted the significance levels obtained by nlme() (ns > 0.05, * 0.01 < $p$ < 0.05, ** 0.001< $p$ < 0.01, *** $p$ < 0.001) and the intercept is Filter paper Water (the relative summary are reported in Table S2).

| Substrate | Watering Treatment | *GP* (%) | $t_{50}$ (Day) | $t_{75}$ (Day) |
|---|---|---|---|---|
| Filter paper | Water | 0.85 | 25.36 | 1.37 |
| *P. undulatum* soil | Water | 0.66 ** | 26.50 ns | 1.96 * |
| *Eucalyptus* soil | Water | 0.88 ns | 25.28 ns | 2.65 *** |
| Filter paper | $L_{EL}$ | 0.27 *** | 23.63 * | 2.36 ** |
| *P. undulatum* soil | $L_{EL}$ | 0.73 *** | 28.14 *** | 1.85 * |
| *Eucalyptus* soil | $L_{EL}$ | 0.71 *** | 25.44 * | 1.84 *** |
| Filter paper | $L_{PG}$ | 0.35 *** | 31.00 *** | 0.92 * |
| *P. undulatum* soil | $L_{PG}$ | 0.66 *** | 27.05 *** | 1.74 ns |
| *Eucalyptus* soil | $L_{PG}$ | 0.86 *** | 24.34 *** | 1.45 * |
| Filter paper | $L_{PL}$ | 0.51 *** | 30.37 *** | 1.40 ns |
| *P. undulatum* soil | $L_{PL}$ | 0.71 *** | 26.50 *** | 1.88 ns |
| *Eucalyptus* soil | $L_{PL}$ | 0.86 *** | 26.33 *** | 2.01 ns |

Regarding the other indices related to the time and duration of the germination (i.e., $t_0$, $t_f$ and $t_f - t_0$), it is possible to observe that the values obtained for all substrates and watering treatments for *E. ovata* seeds, had means in a range of: 4–5.3 days for $t_0$, 11.3–15.7 days for $t_f$, 6.7–11.3 days for $t_f - t_0$ (Table 3). No statistically significant differences were found with glm among the watering treatments and the soils for *E. ovata* seeds for these indices with respect to the reference (i.e., filter paper and water) (Tables 3 and S3–S5). On the other hand, the Vigor Index (VI) showed statistically significant differences between all treatments (i.e., substrates and watering) for *E. ovata* seedlings (Table 3), with the exception of dishes with *P. undulatum* soil and $L_{PG}$ that showed no significant differences. The two-way ANOVA and the following Tukey *post-hoc* test, carried out on this index, which relates to the development of shoot and root tissues, showed that it was influenced by

the substrate and the watering treatment, with the highest values recorded for filter paper watered with distilled water and *P. undulatum* soil watered with $L_{PG}$ (Table 3). Observing the treatments with distilled water, the *Eucalyptus* spp. soil was the substrate with the lowest statistically significant VI values, and the same substrate showed significantly low values with the $L_{PG}$ treatment (Table 3). Observing the filter paper substrate, it is possible to notice that the $L_{EL}$ and $L_{PG}$ are the leachates with significantly lower VI values (Table 3).

**Table 3.** Germination of *Eucalyptus ovata* and *Pittosporum undulatum* seedlings under different substrates (i.e., Filter paper; *P. undulatum* soil; *Eucalyptus* soil) and different watering treatments (i.e., Water; $L_{EL}$ = Leachate of *Eucalyptus* spp. Litter; $L_{PG}$ = Leachate of *P. undulatum* Green leaves; $L_{PL}$ = Leachate of *P. undulatum* Litter). In the $t_0$, $t_f$ and $t_f - t_0$ columns are reported the means ($n$ = 3) and in superscript are noted the significance obtained from the glm model (ns > 0.05, * 0.01 < $p$ < 0.05) and the intercept is Filter paper Water (the relative summaries are reported in Tables S3–S8). Column VI shows the mean ± SD of Vigor Index, while the different letters indicate a significant difference among treatments obtained using the Tukey *post-hoc* test carried out after the two-way ANOVA. All results described as significant were at $p$ < 0.05.

| | | *Eucalyptus ovata* SEEDS | | | | *Pittosporum undulatum* Seeds | | | |
|---|---|---|---|---|---|---|---|---|---|
| **Substrate** | **Watering Treatment** | $t_0$ **(Day)** | $t_f$ **(Day)** | $t_f - t_0$ **(Day)** | **VI (% * mm)** | $t_0$ **(Day)** | $t_f$ **(Day)** | $t_f - t_0$ **(Day)** | **VI (% * mm)** |
| Filter paper | Water | 5.0 | 14.0 | 9.0 | 1390.5 ± 164.4 [a] | 21.3 | 31.0 | 9.7 | 4494.3 ± 126.7 [ab] |
| *P. undulatum* soil | Water | 4.7 [ns] | 11.7 [ns] | 7.0 [ns] | 1310.0 ± 217.3 [ab] | 22.7 [ns] | 32.0 [ns] | 9.3 [ns] | 3272.6 ± 488.1 [b] |
| *Eucalyptus* soil | Water | 4.0 [ns] | 11.3 [ns] | 7.3 [ns] | 876.3 ± 278.9 [bcde] | 20.0 [ns] | 32.0 [ns] | 12.0 [ns] | 5424.3 ± 581.2 [a] |
| Filter paper | $L_{EL}$ | 4.7 [ns] | 14.3 [ns] | 9.7 [ns] | 550.9 ± 140.6 [de] | 19.7 [ns] | 28.7 [ns] | 9.0 [ns] | 1260.3 ± 285.1 [c] |
| *P. undulatum* soil | $L_{EL}$ | 5.3 [ns] | 15.7 [ns] | 10.3 [ns] | 1044.2 ± 93.8 [abc] | 22.3 [ns] | 32.3 [ns] | 10.0 [ns] | 3376.4 ± 312.1 [b] |
| *Eucalyptus* soil | $L_{EL}$ | 4.0 [ns] | 12.3 [ns] | 8.3 [ns] | 1134.1 ± 99.8 [abc] | 19.3 [ns] | 31.0 [ns] | 11.7 [ns] | 3794.9 ± 203.7 [ac] |
| Filter paper | $L_{PG}$ | 4.7 [ns] | 15.6 [ns] | 11.0 [ns] | 446.0 ± 54.9 [e] | 27.3 [ns] | 33.0 [ns] | 5.7 [ns] | 493.3 ± 238.5 [c] |
| *P. undulatum* soil | $L_{PG}$ | 4.7 [ns] | 11.3 [ns] | 6.7 [ns] | 1414.6 ± 18.1 [a] | 22.7 [ns] | 32.7 [ns] | 10.0 [ns] | 3829.2 ± 985.2 [ab] |
| *Eucalyptus* soil | $L_{PG}$ | 4.0 [ns] | 13.0 [ns] | 9.0 [ns] | 723.6 ± 78.1 [cde] | 20.0 [ns] | 31.3 [ns] | 11.3 [ns] | 4525.3 ± 428.9 [ab] |
| Filter paper | $L_{PL}$ | 5.0 [ns] | 15.0 [ns] | 10.0 [ns] | 1179.1 ± 189.4 [ab] | 26.7 [ns] | 32.7 [ns] | 6.0 [ns] | 1282.5 ± 289.1 [c] |
| *P. undulatum* soil | $L_{PL}$ | 4.3 [ns] | 14.0 [ns] | 9.7 [ns] | 927.2 ± 108.8 [bcd] | 22.0 [ns] | 32.0 [ns] | 10.0 [ns] | 3511.6 ± 728.7 [ab] |
| *Eucalyptus* soil | $L_{PL}$ | 4.3 [ns] | 15.7 [ns] | 11.3 [ns] | 1036.9 ± 160.0 [abc] | 21.0 [ns] | 31.0 [ns] | 10.0 [ns] | 3508.9 ± 1622.4 [ab] |

For the indices regarding germination time and duration (i.e., $t_0$, $t_f$, $t_f - t_0$) for *P. undulatum* seeds, calculated with glm, no significant differences were found with respect to the dishes with the substrate of filter paper and distilled water as watering (Tables 3 and S6–S8). Furthermore, the values obtained for all substrates and watering treatments had means in a range of: 19.3–27.3 days for $t_0$, 28.7–33.0 days for $t_f$, 5.7–12.0 days for $t_f - t_0$ (Table 3). Finally, regarding the VI (Table 3), in the dishes with distilled watering treatment, the *P. undulatum* soil substrate resulted in the lowest, while *Eucalyptus* soil substrate presented the highest values. Observing the dishes with filter paper substrate, all treatments with leachates resulted in the lowest values (Table 3). At the end of the experiment, a squash test was carried out and all the ungerminated seeds resulted viable seeds.

### 3.2. Total Saponins and Total Condensed Tannins Content

In the $E_{EG}$ and $E_{EL}$, the amount of saponins were higher than the corresponding *P. undulatum* extracts (i.e., $E_{PG}$ and $E_{PL}$), while no statistical differences were found between $L_{EL}$ and $L_{PL}$ (Figure 4A). Total saponin content was almost 6-fold lower in the $E_{PG}$ ($33.42 \pm 3.32$ mg DE $g^{-1}$ DW) than in $E_{EG}$ ($208.78 \pm 28.56$ mg DE $g^{-1}$ DW) (Figure 4A). In both extracts of *Eucalyptus* spp. (i.e., $E_{EG}$, $E_{EL}$) the content of tannins was higher than that of *P. undulatum* extracts (Figure 4B). In the $L_{EL}$, total condensed tannins were about 10-fold higher ($2.99 \pm 0.55$ mg CE $g^{-1}$ DW) than in the $L_{PL}$ ($0.27 \pm 0.09$ mg CE $g^{-1}$ DW).

### 3.3. HPLC-DAD Analyses of Polyphenol Content

Different polyphenols were identified and quantified in each extract and leachate of *Eucalyptus* spp. and *P. undulatum* (Figures S3–S8, Tables S9–S14). In detail, hydrolysable tannins derived from both ellagic and gallic acids were detected and here identified as

gallic and ellagic acids derivatives. The ellagic acid derivatives were the richest in both $E_{EL}$ and $L_{EL}$, representing more than 90% of the total tannin content in $E_{EL}$ and more than 75% in $L_{EL}$. On the other hand, in $E_{LG}$, the most abundant tannins were gallic acid derivatives (around 65% of total tannins content). Gallic and ellagic acids derivatives were significantly higher in $E_{EL}$ and $L_{EL}$ compared to the $E_{EG}$ and to the *P. undulatum* extracts and leachate (Figure 5A). The myricetin derivatives were the only flavonoids identified in the extracts of *Eucalyptus* and *P. undulatum*. Considering the sum of all the myricetin derivatives, here reported as total flavonoid content, it was higher in the $E_{EG}$ and $E_{EL}$ compared to the $E_{PG}$ and $E_{PL}$. In addition, these compounds were very low in the $L_{PL}$ and not detected in the $L_{EL}$ (Figure 5B). Caffeic and p-coumaric acids derivatives (here reported as total hydroxycinnamic acid derivatives, Figure 5C) were identified only in *P. undulatum* extracts and leachate. The highest value of hydroxycinnamic acid derivatives was found in the $E_{PL}$ (2.40 ± 0.29 mg g$^{-1}$ of DW).

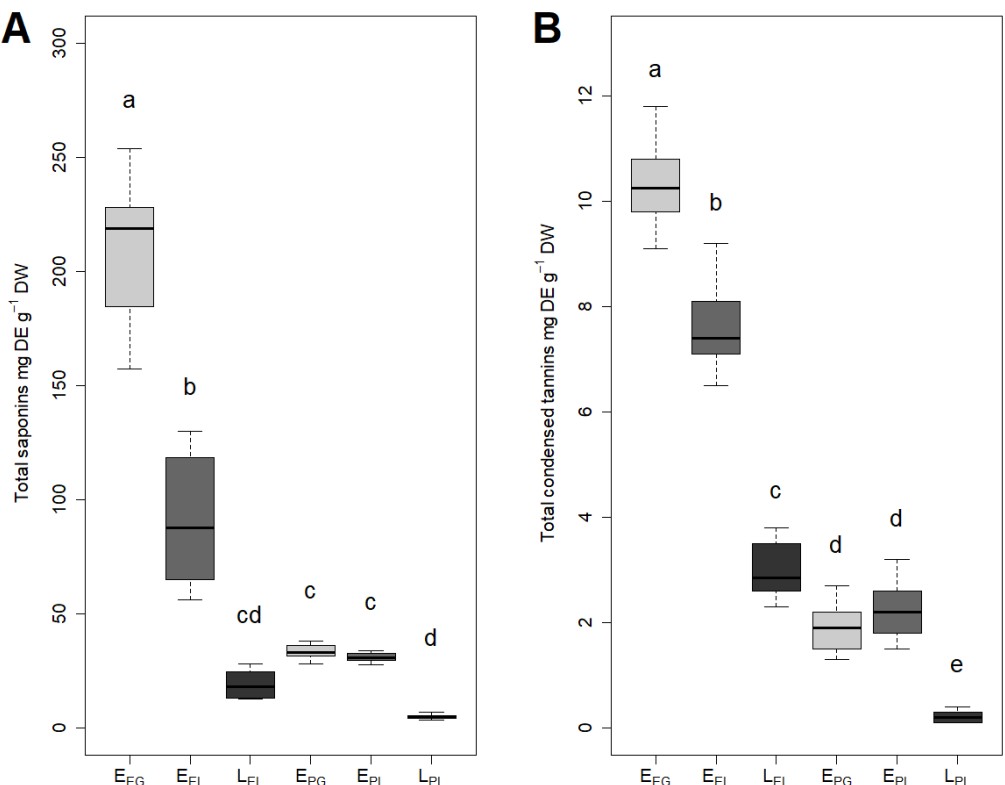

**Figure 4.** (**A**) Content of total saponin, expressed as mg of Diosgenin Equivalent (DE) g$^{-1}$ DW, (**B**) content of total condensed tannins, expressed as mg of Catechin Equivalent (CE) g$^{-1}$ DW, in *Eucalyptus* spp. and *P. undulatum* extracts and leachates (i.e., $E_{EG}$ = Extract of *Eucalyptus* spp. Green leaves; $E_{EL}$ = Extract of *Eucalyptus* spp. Litter; $L_{EL}$ = Leachate of *Eucalyptus* spp. Litter; $E_{PG}$ = Extract of *P. undulatum* Green leaves; $E_{PL}$ = Extract of *P. undulatum* Litter; $L_{PL}$ = Leachate of *P. undulatum* Litter). Each value is the mean of three measurements ± SD. Bars with the same letter are not significantly different when analysed by a one-way non-parametric analysis of variance (Kruskal–Wallis Test) followed by a Dunn's Multiple Comparison post-hoc test.

### 3.4. BVOC Analysis

More than 80% of BVOCs identified in the area invaded by *P. undulatum* (I) are represented by MTs and only 20% by SQTs. In the area of natural vegetation characterised by the presence of *Eucalyptus* spp. and *Acacia* spp. (R), the percentages of MTs and SQTs are more similar to each other: 55% of the BVOCs identified are represented by MTs and 45% by SQTs. The relative concentrations of monoterpenes and monoterpenoids found in the two studied areas are shown in Figure 6. Results from one-way ANOVA test showed that all the identified MTs are significantly different between the two sites, except for the

molecules annotated as α-phellandrene, 1,8-cineole and terpinyl acetate. Additionally, it is possible to note that all MTs, excepting the 1,8-cineole, are higher in the area invaded by *P. undulatum* (I) compared to the air collected in the remnant area (R). The percentages of sesquiterpenes and sesquiterpenoids identified in the two studied areas are reported in Figure 7. All SQTs (excepted the compound annotated as selinadiene) are significantly higher in the remnant vegetation compared to the area invaded by *P. undulatum*, where calamenene is the only SQTs significantly higher.

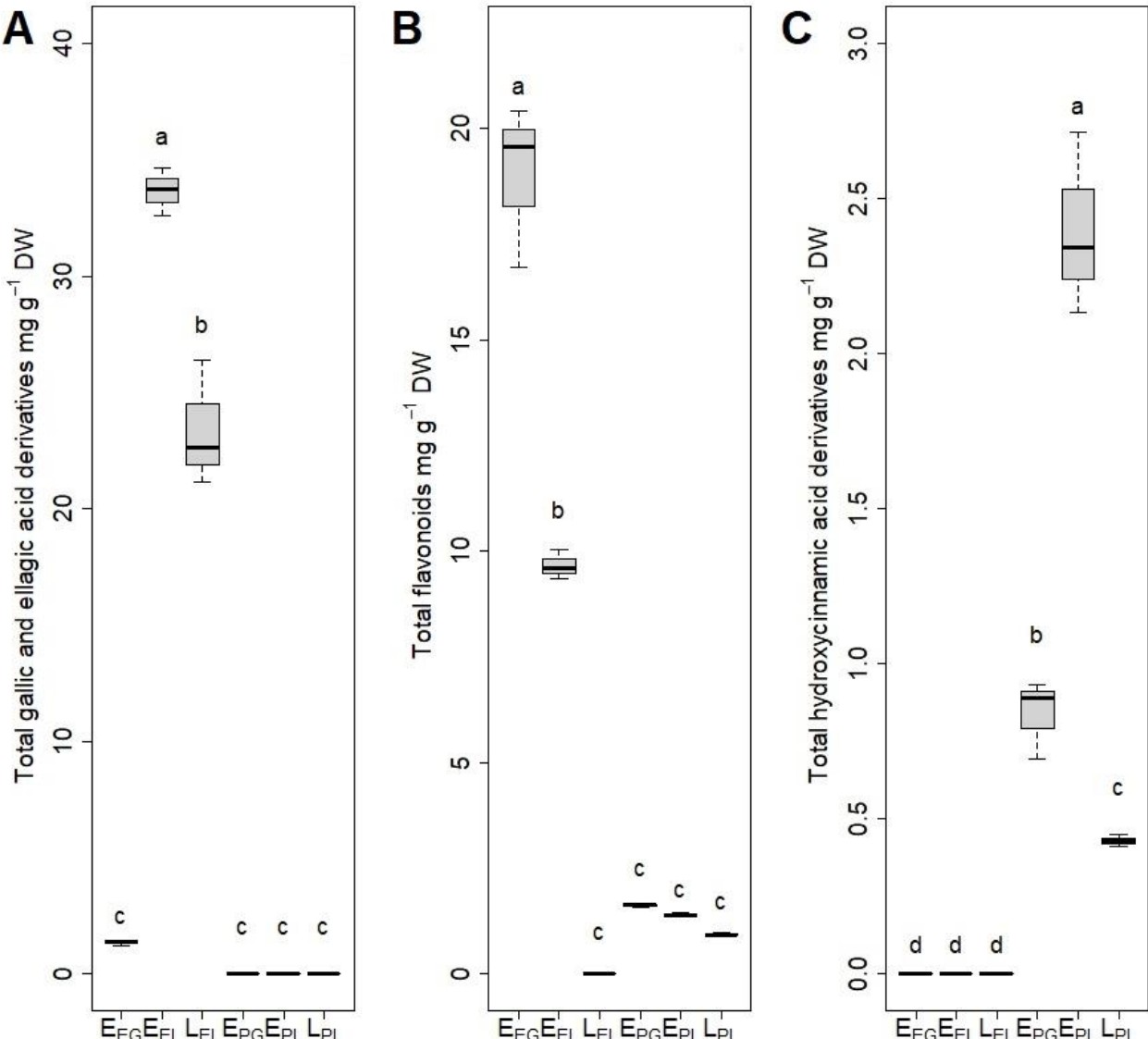

**Figure 5.** Total content of gallic and ellagic acids derivatives (**A**), Total Flavonoid Content (**B**) and Total Hydroxycinnamic Acid derivatives (**C**) of *Eucalyptus* and *P. undulatum* extracts and leachates (mg g$^{-1}$ DW) (i.e., $E_{EG}$ = Extract of *Eucalyptus* spp. Green leaves; $E_{EL}$ = Extract of *Eucalyptus* spp. Litter; $L_{EL}$ = Leachate of *Eucalyptus* spp. Litter; $E_{PG}$ = Extract of *P. undulatum* Green leaves; $E_{PL}$ = Extract of *P. undulatum* Litter; $L_{PL}$ = Leachate of *P. undulatum* Litter). Each value is the mean of three measurements ± SD. Bars with the same letter are not significantly different when analysed by a one-way non-parametric analysis of variance (Kruskal–Wallis Test) followed by a Dunn's Multiple Comparison post-hoc test.

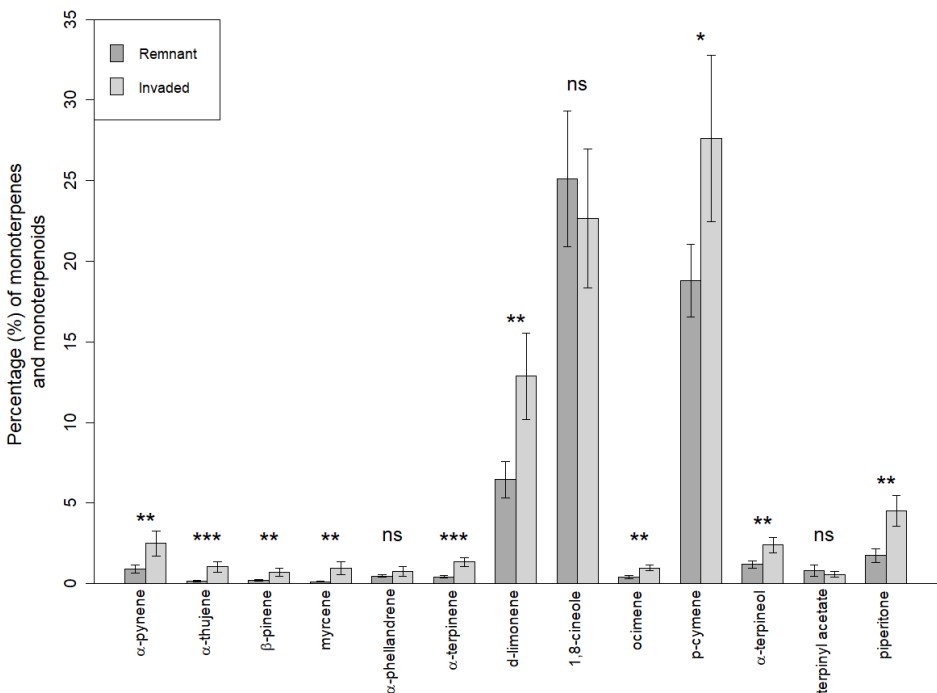

**Figure 6.** Histogram representing the amount (%) of monoterpenes and monoterpenoids annotated as volatiles collected in remnant area of natural vegetation (R) and areas invaded by *P. undulatum* (I). Each value is the mean of five measurements ± SD. Error bars indicate standard deviation, and the asterisks indicate significant differences between R and I: ns > 0.05, * 0.01 < *p* < 0.05, ** 0.001 < *p* < 0.01, *** *p* < 0.001.

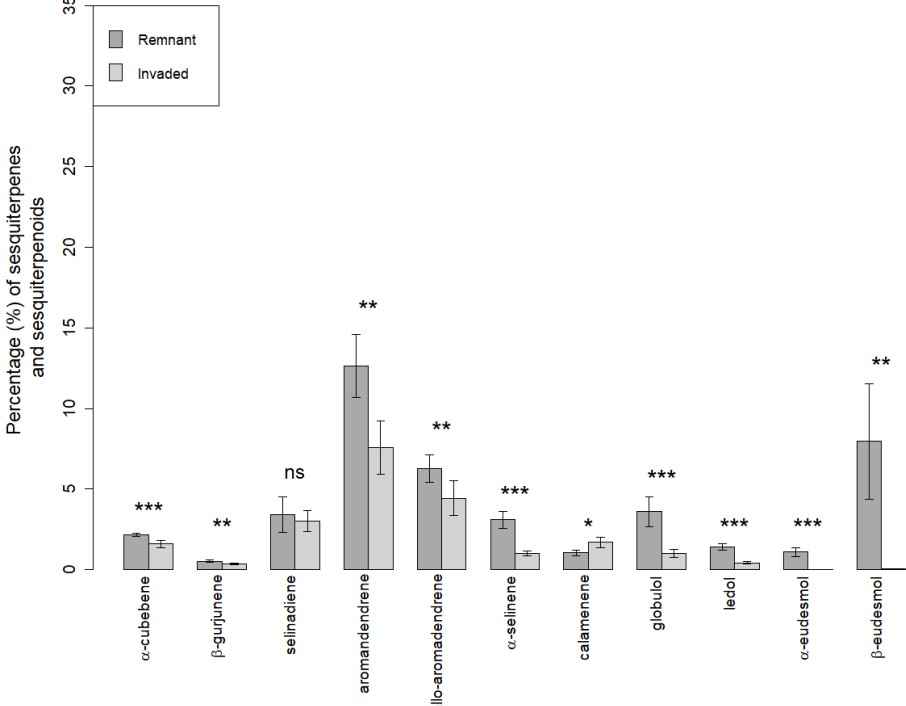

**Figure 7.** Histogram representing the amount (%) of sesquiterpenes and sesquiterpenoids annotated as volatiles collected in remnant area of natural vegetation (R) and areas invaded by *P. undulatum* (I). Each value is the mean of five measurements ± SD. Error bars indicate standard deviation, and the asterisks indicate significant differences between R and I: ns > 0.05, * 0.01 < *p* < 0.05, ** 0.001 < *p* < 0.01, *** *p* < 0.001.

## 4. Discussion

### 4.1. Does Pittosporum undulatum Inhibit Germination of Eucalypts?

Several aspects of germination were examined (percentage, time to 50% germination, and Vigor Index). There was very little evidence for total suppression of germination, but there were significant differences in the rate of germination, particularly in seeds grown on filter paper over soil. There was also evidence that leachates may be affect the growth of young shoots and roots. However, overall, our results do not definitively link *P. undulatum* invasiveness to allelopathic actions against *E. ovata* seeds.

The germination curves indicated the absence of total suppression of seed germination due to *Pittosporum undulatum* and *Eucalyptus* spp. leachates and/or soil. Additionally, for *E. ovata* seeds, no statistical difference was detected as a result of the watering treatments, which did not show any statistical influence, resulting in its removal from the model. The lowest values of germinated percentage seeds (*GP*) was in the dishes with soil collected under *Eucalyptus* spp. plants, in accordance with the work of Zhang and Fu [53], who showed high allelopathic action of *Eucalyptus* spp. from the roots but not from the litter alone. Instead, the $t_{50}$ and the $t_{75}$ parameters showed significant differences between soil substrates (i.e., *P. undulatum* and *Eucalyptus* soil) with respect to filter paper, with lower time (days) required to reach the 50% of the total final germination and a more rapid growth during the exponential phase.

The longer time recorded to reach 50% germination in Petri dishes with filter paper compared to the soil substrate may be due to the greater aeration and infiltration of the leachate into the soil substrate compared to filter paper, where the liquid can remain more in contact with the seed, extending its dormancy [40]. In addition, the absence of significant differences from the indices regarding germination time and duration (i.e., $t_0$, $t_f$ and $t_f - t_0$) allowed us to assume that the *P. undulatum* leachates (i.e., $L_{PG}$ and $L_{PL}$), in all substrates (i.e., filter paper, soil collected under *P. undulatum* plants and *Eucalyptus* spp. plants), did not show any allelopathic action affecting the germination time of *E. ovata* seeds. Observing the Vigor Index (VI) of *E. ovata* seedlings, a similar assumption regarding a high allelopathic action from *Eucalyptus* roots could be made for the dishes with distilled water treatment, which showed lower VI values in the dishes with soil collected under *E.* spp. plants. Moreover, an allelopathic action from the $L_{PG}$ treatment, in the dishes with filter paper substrate and soil collected under *Eucalyptus* plants, showed more damages caused by *P. undulatum* leachate against young tissues of shoots and roots than a germination inhibition. Finally, the lowest values were observed in the filter paper substrate, indicating a possible chelating action of the soil. A possible soil chelating action is also strongly evident when observing the results from Germination Percentage (*GP*), time of germination (especially $t_{50}$ and $t_0$) estimates from *Pittosporum undulatum* seeds. Indeed, for *GP* parameter, the dishes with the lowest statistically significant differences were those with filter paper substrates and leachate treatments (i.e., $L_{EL}$, $L_{PG}$, and $L_{EL}$).

Regarding $t_{50}$ and $t_0$, the dishes that required a longer time to start germinating and to reach 50% of the total germination were $L_{PG}$ and $L_{PL}$ in filter paper substrate. Indeed, it is important to note that the soil is a very complex system, which can influence the qualitative and quantitative availability of phenolics and other allelopathic compounds [54]. When evaluating the possible allelopathic action of some plants, it is necessary to consider the physiochemical properties of the soil, and, thus, when carrying out allelopathic tests of germinability, it is important to use the same soils that plants find in nature [55]. In the literature, phytotoxic effects of *P. undulatum* leaf extracts on the germination of native Australian species varied considerably. Gleadow and Ashton [14] carried out a germination test, where they reported an allelopathic action of this invasive species that could explain its rapid spread: *P. undulatum* leaves leachate suppressed germination in *E. obliqua* (47%), in *E. melliodora* (8%) and in *E. goniocalyx* (48%). Our results disagree with these previous results; however, the absence of allelopathic action from *P. undulatum* leaf leachate is in accordance with Tunbridge et al. [56], which reported a significant increment of germination rates of *E. viminalis* when seeds were treated with *P. undulatum* leaf leachate (+70%) compared

to untreated seeds (watering with distilled water). These results could be explained by the fact that allelopathic actions of *P. undulatum* could differ depending on different species and subspecies exposed to them. Moreover, regarding our result of inhibition of *P. undulatum* leaf leachate on *P. undulatum* seeds germination (in the dishes with filter paper substrate and $L_{PG}$ and $L_{PL}$ treatments) and of the reduction in seedlings development (regarding VI in the dishes with $L_{PG}$ and $L_{PL}$), other authors reported an inhibitory action on *P. undulatum* seedlings under mature *P. undulatum* canopy [57]. Finally, our data indicated an allelopathic action of *Eucalyptus* spp. litter leachates on *P. undulatum* seed germination and on *P. undulatum* and *E. ovata* development of seedlings (VI). This result is in accordance with the literature. Indeed, previous studies have shown that many eucalypt species present allelopathic actions, which may inhibit and suppress seed germination and seedling establishment of other species [58,59].

As suggested by the germination test results, the lack of allelopathic action of *P. undulatum* on *Eucalyptus ovata* seeds might be correlated with the low values of saponins and tannins content found in its leachate. Indeed, the phytotoxicity of saponins and tannins has been linked to a general reduction in the growth of seedlings and to the inhibition of germination in the exposed plant organisms. On the other hand, the higher content of saponins in eucalypt extracts, concomitantly with a higher tannin content with respect to the corresponding extracts obtained from *P. undulatum*, suggests a major investment of carbon for the biosynthesis of these secondary metabolites in *Eucalyptus* to improve defence against biotic stresses [60]. Indeed, as previously reported in the literature, for other species [32,61], saponins and tannins may help in protecting leaves against insects and herbivores by decreasing the digestibility of their tissues [62,63].

Regarding the possible allelopathic actions of volatile compounds, in the literature it is reported that both monoterpenes and sesquiterpenes are known as allelochemical substances [35,64]. Among the monoterpenes and monoterpenoids, the cineoles (1,8-cineole and 1,4-cineole) are considered the most inhibitory terpenes, followed by camphor, citronellol, menthol, and linalool [65–68]. Indeed, 1,8-cineole may block the mitosis, damaging seedlings growth, inhibit respiration, and increase membrane permeability, causing cell destruction [69]. Among the monoterpenes listed above, only the 1,8-cineole was detected in the two areas during our study, and it was more abundant in the area without *P. undulatum,* although not statistically significant. While, with regards to the allelopathic activity of sesquiterpenes [11], sesquiterpene lactones [70] and β-caryophyllene [35] have been reported to strongly inhibit seed germination and root growth. All these sesquiterpenes were not identified in our study. These results further show that in the area invaded by *P. undulatum*, no volatile allelopathic compounds were detected.

*4.2. Is the Invasiveness of P. undulatum due to the Storing and Emission of Secondary Metabolites?*

Our data support the hypothesis that the high invasiveness of *P. undulatum* may be due to the storing and emission of secondary metabolites. Different polyphenol classes were found in the *Eucalyptus* spp. and *P. undulatum* extracts and leachates. Eucalypt extracts and leachate are richer in tannins, and their biosynthesis is induced by the interactions between plants and herbivores [71], as well as in response to environmental stresses, since tannins may also play an antioxidant role [72]. By contrast, *P. undulatum* extracts and leachate are characterised by higher contents of hydroxycinnamic acid derivatives, with caffeic acid derivatives as the most abundant compounds, in agreement with findings of Nunes et al. [19]. The role of caffeic acid in cell plants may be linked to their function against abiotic stresses [73]. Indeed, caffeic acid is utilized by plants to synthesise lignin to increase the thickness of cell walls as a response to salinity [74], drought stress [73], and intense light [75]. Furthermore, caffeic acid derivatives have a stable structure to trap free radicals, thus playing an antioxidant role against reactive oxygen species (ROS) [76]. Therefore, the secondary metabolites identified in the leachate and leaf extracts of *Pittosporum undulatum* (i.e., hydroxycinnamic acid derivatives) do not seem to play an allelopathic role, as initially hypothesized, but have been associated with a greater defense by this species against abiotic

stresses. Instead, the secondary metabolites (i.e., saponins, condensed tannins, gallic- and ellagic-acid derivatives) identified in the leachate and in the *Eucalyptus* spp. extracts, show how this species accumulates substances that make its leaves more bitter and more difficult to digest, in order to defend itself from possible herbivore attacks.

Regarding terpenes, the forest invaded by *P. undulatum* had a higher percentage of monoterpenes. These compounds play an important role in the defence against abiotic and biotic stresses: thermo tolerance under heat waves [77,78], high light tolerance [64], counteracting the production of ROS in response to drought [79], attracting natural enemies of herbivores [80], and attracting pollinators [81]. By contrast, it is important to note that 1,8-cineole is one of the main components of the *Eucalyptus* spp. terpenic profile [82], which is reflected by its higher content in the remnant area (R), although this is not statistically significant. Unlike other monoterpenes, this compound has been linked to a lower efficiency in leaf thermal protection [83], while its principal role is likely the regulation of plant–insect interactions, such as attracting pollinators [84], repelling herbivores [63], and providing antimicrobial properties [85,86]. Regarding the SQTs content, whose amount was found higher in the remnant area, several studies reported that the sesquiterpene roles are mainly related to plant biotic interactions, since they are semivolatile compounds and principally act as repellents against herbivorous insects [64,87,88].]

The difference between the terpenes in the two types of vegetation can be attributable to their different tree compositions. Indeed, it is important to notice that the sampling sites are natural and complex environments, in which other plants emitting MTs could also be found. Indeed, in the invaded sampling site (I), in addition to *Eucalyptus* spp. (55%) and *P. undulatum* (40%) trees, other species such as acacia (5%), and especially an herbaceous layer, was observed. Instead, in the eucalypt remnant area (R), *Eucalyptus* spp. represented almost the totality of species (95%–98%), with only some individuals of *Acacia* spp. (5%–2%), but neither *P. undulatum* nor grasses were present in this area. The amount of acacia trees was similar in the two areas; however, their possible interference was likely negligible, since this species is considered a low terpenes emitter [89]. Furthermore, in the case of the herbaceous layer found in area I, several studies examining BVOC emissions from grasses found that only light-weight oxygenated BVOCs are emitted [90], with lower emissions compared to trees and shrubs [91]. A potential additional source of BVOCs in area R could be the emissions of soil microbes, but BVOC emission rates from soil microbes in several forest soils are usually very low compared to other sources [92].

We hypothesize that the compounds found in this study derived from *Eucalyptus* and *Pittosporum undulatum* canopies, and the difference between the two areas can be mainly attributed to the presence/absence of *P. undulatum* trees. Furthermore, it is important to remember that plants' terpenic emissions are affected by a genetic component (constitutive terpenes) and by a component induced by abiotic and biotic stresses. In the case of MTs, their emission is largely induced by environmental stresses, especially drought, high temperatures, salinity, and light stress. The two areas were selected near each other (around 700 m away), share the same altitude, and the sampling was conducted at the same time during the same day in both areas, thus, the emission was characterised by the same climatic conditions. The terpenic profile obtained in the area invaded by *P. undulatum* could suggest that monoterpenes are mainly emitted by this species. This is in agreement with the literature reporting that *P. undulatum* is a large emitter of monoterpenes, rather than sesquiterpenes, and that the main terpene is limonene, as also shown in our results [93]. Furthermore, as these monoterpenes are known to improve tolerance to abiotic stresses in several plants [83,94], this allows us to support the hypothesis that the invasiveness of *P. undulatum* may be linked to compounds that play a protective role against climate change. The differences in the terpene concentrations, together with the higher content of saponins and tannins in *Eucalyptus* spp. leaves, could support the hypothesis that eucalypts invest more carbon in defence mechanisms against herbivore attacks, while *P. undulatum* invests in secondary metabolites that may increase its tolerance against abiotic stresses, such as drought and heat. Nevertheless, the lack of a *P. undulatum* control site, with 90%

cover, cannot exclude the occurrence of different monoterpenes profile, different from the one observed in the presence of *Eucalyptus* spp. in the invaded plot. Indeed, Gleadow and Rowan [21] have shown that *P. undulatum* is drought tolerant rather than drought avoiding, since severely wilted leaves of seedlings of *P. undulatum* can regain their turgidity after rewatering, while this was not observed in that study for *Eucalyptus viminalis* seedlings.

## 5. Conclusions

Our study investigated whether the high invasiveness of *Pittosporum undulatum* in *Eucalyptus* forests may be related to the biosynthesis and emission of secondary metabolites, which could have allelopathic actions or increase their defence against abiotic and biotic stresses. Our results show that the characteristic of *P. undulatum* which may confer this species a high invasiveness is the biosynthesis of hydroxycinnamic acid derivatives and monoterpenes, which increase its tolerance against abiotic stresses rather than acting as allelopathic compounds. The secondary metabolites identified in *Eucalyptus* spp. (i.e., higher amounts of gallic and ellagic acid derivatives, flavonoids, and sesquiterpenes) suggest that this species invests in repellent and low-digestible compounds that help prevent herbivores attacks. In addition, the characterisation of the BVOCs found under canopy at the environmental level allowed us to obtain a broader picture of the terpenes emitted in the air, instead of single measurements of each plant.

**Supplementary Materials:** The following supporting information can be downloaded at: https://www.mdpi.com/article/10.3390/f14010039/s1, Figure S1: Temporal progression of seeds germination for each Petri dish ($n$ = 36) for *Eucalyptus ovata* seeds; Table S1: Summary of Nonlinear Mixed-Effects Models for *Eucalyptus ovata* seeds reported in Table 1; Figure S2: Temporal progression of seeds germination for each Petri dish ($n$ = 36) for *Pittosporum undulatum* seeds; Table S2: Summary of Nonlinear Mixed-Effects Models for *Pittosporum undulatum* seeds reported in Table 2; Table S3: Summary of Generalized Linear Models for the time for the first germination ($t_0$, day) of *Eucalyptus ovata* seedlings under different substrates and different watering treatments; Table S4: Summary of Generalized Linear Models for the time for the last germination ($t_f$, day) of *Eucalyptus ovata* seeds under different substrates and different watering treatments; Table S5: Summary of Generalized Linear Models for the time spread of germination ($t_f - t_0$, day) of *Eucalyptus ovata* seeds under different substrates and different watering treatments; Table S6: Summary of Generalized Linear Models for the time for the first germination ($t_0$, day) of *Pittosporum undulatum* seeds under different substrates and different watering treatments; Table S7: Summary of Generalized Linear Models for the time for the last germination ($t_f$, day) of *Pittosporum undulatum* seeds under different substrates and different watering treatments; Table S8: Summary of Generalized Linear Models for the time spread of germination ($t_f - t_0$, day) of *Pittosporum undulatum* seeds under different substrates and different watering treatments; Figure S3: Representative HPLC-DAD chromatogram (at 280 nm) of ethanolic extracts of *Eucalyptus* green leaves; Table S9: List of the compounds detected in ethanolic extracts of green leaves of *Eucalyptus* spp.; Figure S4: Representative HPLC-DAD chromatogram (at 280 nm) of ethanolic extract of *Eucalyptus* litter; Table S10: List of the compounds detected in ethanolic extracts of *Eucalyptus* spp. litter; Figure S5: Representative HPLC-DAD chromatogram (at 330 nm) of ethanolic extracts od *Pittosporum undulatum* green leaves; Table S11: List of the compounds detected in ethanolic extracts of green leaves of *Pittosporum undulatum*; Figure S6: Representative HPLC-DAD chromatogram (at 330 nm) of ethanolic extracts of *Pittosporum undulatum* litter; Table S12: List of the compounds detected in ethanolic extracts of litter of *Pittosporum undulatum*; Figure S7: Representative HPLC-DAD chromatogram (at 280 nm) of aqueous extracts of *Eucalyptus* litter; Table S13: List of the compounds detected in aqueous extracts of litter of *Eucalyptus* spp.; Figure S8: Representative HPLC-DAD chromatogram (at 330 nm) of aqueous extracts of *P. undulatum*; Table S14: List of the compounds detected in aqueous extracts of litter of *P. undulatum*.

**Author Contributions:** Conceptualization, D.P., C.B. and R.M.G.; methodology, D.P., C.B. and R.M.G.; software, D.P.; validation, D.P. and L.B.d.S.N.; formal analysis, D.P. and L.B.d.S.N.; investigation, D.P., C.B. and R.M.G.; resources, C.B., F.F. and R.M.G.; data curation, D.P.; writing—original draft preparation, D.P.; writing—review and editing, D.P., L.B.d.S.N., C.B., F.F. and R.M.G.; visualization, D.P.; supervision, C.B., F.F. and R.M.G.; project administration, R.M.G.; funding acquisition, F.F. and R.M.G. All authors have read and agreed to the published version of the manuscript.

**Funding:** This research received no external funding.

**Data Availability Statement:** Not applicable.

**Acknowledgments:** The authors would like to thank Ben O'Leary for his important assistance with the field work.

**Conflicts of Interest:** The authors declare no conflict of interest.

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
