# Peer review of "Is the Invasiveness of Pittosporum undulatum in Eucalypt Forests Explained by the Wide Ranging Effects of Its Secondary Metabolites?"

_forests, doi:10.3390/f14010039_

Round 1

Reviewer 1 Report (Previous Reviewer 1)

The paper undoubtedly improved from its latest version. The authors answered all my comments addressed. The authors made a significant effort, sometimes overextending the text. The authors better explained that the monoterpenes profile may be ascribed to the presence of P. undulatum, but I repeat that caution must be used as the experimental design lacks the P. undulatum control, i.e., a plot with at least 90% cover from this species. In L721 a sentence should be inserted asserting that the lack of a P. undulatum control with 90% cover cannot exclude the occurrence of a monoterpenes profile different form that observed in presence of Eucalyptus spp. in the invaded plot.

The authors made a huge effort for data analyses, but the presentation is still not clear. For example, in L334-352 what does it mean the symbol ~ or |in Eq. 2.3 and 2.4? If they are only descriptive, they make confusion; write them between brackets or after a semicolon.

As not all readers are familiar with nonlinear mixed-effects model, it is necessary an effort to make the description of the statistic adopted more readable. For example, it should be better explained what it means “the model space was investigated by comparing marginal models”, as you remove the treatment factor for Eucalyptus ovata seeds from the data presentation.  

Another drawback is the verbose style still occurring throughout in the text; it must be shortened. In my opinion the paper is acceptable with minor but significant revisions, taking into account comments which I hope will improve the significance of the manuscript.

L62. The sentence “that has a high invading…” is out of context and without capital letter at the beginning.

L95. You are introducing new concepts, “Furthermore” is not appropriate.

L101. To refer to all the categories of secondary metabolites seems that all of them have allelophatic activity. It should be correct to write “… of several allelopathic compounds belonging to the … classes…”

L208 delete “with”. Moreover, move “giving a total of 72 Petri dishes and 2160 seeds” from L208 to L206 after “… for each combination” in order to improve the readability.

L209. What was the light intensity (in PAR)?

L218-233. The use of R words does not help the readability. Write the parameter you estimated, i.e., GP, t50, t75 (I suggest the use of this word for the ¾ of GP). The names of R glossary must be reported within brackets at the end of the sentence for each parameter.

L372. Why do you use the Dunnett test? It is for parametric distribution, but you stated it is not. Moreover, Dunnett test compare all treatments against a control. Dunn's Multiple Comparison Test is a post hoc non parametric test.

Table 2 and 3. Grouping by substrate or by water solutions (1st or 2nd column) should improve the readability. As it is now, the first three are grouped by water solution, the rest is mixed.

Figure 3. Panel A can be removed; its readability is poor and adds no information with respect to panels B and C.

Figure 4 and 5 are very large; may be assembled as panels of the same figure, like Figure 6.

Figure 8. Substitute the text in the caption with sesquiterpenes and sesquiterpenoids in place of monoterpenes and monoterpenoids

Author Response

We thank the reviewer for his/her useful comments. 

We have added a sentence regarding the missing P. undulatum control in paragraph 4.2, as suggested. We also clarified the text regarding the nonlinear mixed-effect model. Lastly, we decided to keep the R language formula format for ease of replication by readers.

-L62. The sentence “that has a high invading…” is out of context and without capital letter at the beginning.

We have corrected the typo

-L95. You are introducing new concepts, “Furthermore” is not appropriate.

We have removed “furthermore”. We have rewritten this section of the introduction (lines 96-104).

-L101. To refer to all the categories of secondary metabolites seems that all of them have allelophatic activity. It should be correct to write “… of several allelopathic compounds belonging to the … classes…”

Thank you for your suggestion, we have modified the text following your comment.

-L208 delete “with”. Moreover, move “giving a total of 72 Petri dishes and 2160 seeds” from L208 to L206 after “… for each combination” in order to improve the readability.

Thank you for the comment. We have now modified the text as follows: “… Thus, the experimental design consisted of 12 combinations (3 substrates × 4 watering treatments) per species and three replicates for each combination, giving a total of 72 Petri dishes (90 mm, Filter-paper Advantech type 2) and 2160 seeds (30 seeds per dish). Each replicate consisted of a Petri dish….” (Lines 209-214)

-L209. What was the light intensity (in PAR)? 

The cabinet had a light intensity of 200 µM m-2 s-1 PAR. This detail has been added to the text.

-L218-233. The use of R words does not help the readability. Write the parameter you estimated, i.e., GP, t , t (I suggest the use of this word for the ¾ of GP). The names of R glossary must be reported within brackets at the end of the sentence for each parameter.

Thank you for the comment. We have now changed the text as suggested.

-L372. Why do you use the Dunnett test? It is for parametric distribution, but you stated it is not. Moreover, Dunnett test compare all treatments against a control. Dunn's Multiple Comparison Test is a post hoc non parametric test.

Thank you for the comment, and sorry for the confusion. We had actually used the Dunn’s Multiple Comparison Test in R, but mistakenly reported it as Dunnet. We have now corrected the text. Thank you for picking it up.

-Table 2 and 3. Grouping by substrate or by water solutions (1 or 2 column) should improve the readability. As it is now, the first three are grouped by water solution, the rest is mixed.

We have decided to keep the same order in tables 2 and 3, following the order output from R summary, which is also reported in the tables S2-S8 in the supplementary materials. In the new version of the manuscript we have  modified table 2 and 3 and tables S2-S8 in the supplementary following your suggestion.

-Figure 3. Panel A can be removed; its readability is poor and adds no information with respect to panels B and C.

We have modified Figure 3 as suggested.

-Figure 4 and 5 are very large; may be assembled as panels of the same figure, like Figure 6.

We have modified Figure 4 and 5 as suggested.

-Figure 8. Substitute the text in the caption with sesquiterpenes and sesquiterpenoids in place of monoterpenes and monoterpenoids

We have modified Figure 8 as suggested.

Reviewer 2 Report (Previous Reviewer 2)

Authors have made important and significant changes in this new amended version. I am totally satisfied with this new version and recommend the paper for publication. I only have two final suggestions:

1) Authors now state an interesting question in their title. I recommend to match this question with what is said in Abstract (L17-19). The title talks about secondary metabolites (SMs) but the Abstract talks about SMs plus allelopathic compounds. They must coincide. I also suggest to providing an answer to this question in the Conclusion section.

2) Please carefully check for typos and small mistakes in the text. A few examples: L62 (dot after Victoria and double space after the word ‘its’), L389 (Observing and observe).

Author Response

We thank the reviewer for his/her comments.  We have now modified the abstract and the conclusion following the suggestions and we have carefully checked the text and removed all typos. We have also made some minor changes in the text  to improve clarity and flow.

This manuscript is a resubmission of an earlier submission. The following is a list of the peer review reports and author responses from that submission.

Round 1

Reviewer 1 Report

The paper is very interesting and address the invasiveness of a species at different scales, from germinability to BVOCs emission. The research is well presented, but the Discussion lacks a clear interpretation of the obtained results from what is known. In particular, caution must be used as the experimental design lacks the P. undulatum control, i.e., a plot with at least 90% cover from this species. Inferring that the higher concentrations of monoterpenes in the invaded plot is of P. undulatum origin is reasonable, but the cover of 40% and the lack of the P. undulatum control cannot exclude a specific response to the presence of Eucalyptus in the invaded plot. It is advisable to rewrite the Discussion focusing on the presented data. English needs an extended revision; it is verbose particularly in the Discussion section. In my opinion the paper is not acceptable in the present version and needs major revision, but I strongly encourage the authors to resubmit an improved version of this manuscript that takes into account comments which I hope will improve the significance of the manuscript.

L77. Replace “believe” with “consider”, sounds more scientific

L80. Delete the dash at the end

L139. Probably “The small seeds are contained…”

L218. Total saponins and condensed tannins contents were…

Moreover, extracts definitions are inconsistent with other parts of the manuscript (L307-309). I suggest: “… on litter water leachates… and… spp. litter and green leaves extracts prepared…”.

L225. total saponin content (TSC) – the acronym written when first mentioned. The same at L233: total condensed tannin content (TCT). Query: having incorporated the term “content” in the saponin acronym, the adoption of the same criterion for tannins should be more coherent, i.e., TTC (Total condensed Tannins Content).

L321. In my opinion, it is unusual to write an affirmative sentence in the heading.

Figure 2 and 3. Is it possible to arrange in a single figure all the six panels? Standardizing the X-axis scale (35 days should be enough) improve the comparison/readability.

Table 1 and 2. You write in L303 that ANOVA was conducted between watering treatments for each substrate, separately, but lowercase letters do not seem to follow this criterion. For example, in Table 1, filter-paper substrate, t50 (Farooq) has b for all four watering treatments.

Figure 7. Being percentage, standardize the Y-axis for both panel at 35% should improve the comparison/readability.

L511. “… in the growth of seedlings and to the inhibition of germination” of what? Its own or other species?

L523. What the bicyclic sesquiterpenoid refer to: the caryophyllene or lactones?

L538. functional?

L542. creating?

L547. tended to higher. Simply “are higher” sounds better

Author Response

We thank the reviewer for the useful comments.  We have modified and expanded the discussion accordingly with the suggestions of the reviewer. Here we provide a point-by-point response and a tracked-changes version of the manuscript (changes marked in red). 

Comment: L77. Replace “believe” with “consider”, sounds more scientific

Answer: Thank you for your suggestion. We have modified the verb

Comment: L80. Delete the dash at the end

Answer: Thank you for your comment. We have corrected the typo

Comment: L139. Probably “The small seeds are contained…”

Answer: Thank you for your comment. This sentence has been removed during rewording of the manuscript

Comment: L218. Total saponins and condensed tannins contents were…

Answer: Thank you for your comment. We have modified the text [now L245]

Comment: Moreover, extracts definitions are inconsistent with other parts of the manuscript (L307-309). I suggest: “… on litter water leachates… and… spp. litter and green leaves extracts prepared…”.

Answer: Thank you for your suggestion. We have modified the text and the figures according to your comment opting for acronyms.

Comment: L225. total saponin content (TSC) – the acronym written when first mentioned. The same at L233: total condensed tannin content (TCT). Query: having incorporated the term “content” in the saponin acronym, the adoption of the same criterion for tannins should be more coherent, i.e., TTC (Total condensed Tannins Content).

Answer: Thank you for your comment and suggestion. We have corrected the lines and replace “TCT” with “TcTC” in the text [now L262]

Comment: L321. In my opinion, it is unusual to write an affirmative sentence in the heading.

Answer: Thank you for your comment, we have modified the sub-headings in the Results section [now L381]

Comment: Figure 2 and 3. Is it possible to arrange in a single figure all the six panels? Standardizing the X-axis scale (35 days should be enough) improve the comparison/readability.

Answer: Thank you for the suggestion. We now modified the whole section 3.1 according to the requests of the other reviewer

Comment: Table 1 and 2. You write in L303 that ANOVA was conducted between watering treatments for each substrate, separately, but lowercase letters do not seem to follow this criterion. For example, in Table 1, filter-paper substrate, t50 (Farooq) has b for all four watering treatments.

Answer: There was an error in the materials and methods. We have now changed the statistical analyses regarding the germination experiment.  

Comment: Figure 7. Being percentage, standardize the Y-axis for both panel at 35% should improve the comparison/readability.

Answer: Thank you for your suggestion. We have modified the y-axis of figure 7 and 8

Comment: L511. “… in the growth of seedlings and to the inhibition of germination” of what? Its own or other species?

Answer: Thank you for your comment. This reduction was relative to the plants exposed to such compounds. We have modified the text to make it clearer [now L636-637]

Comment: L523. What the bicyclic sesquiterpenoid refer to: the caryophyllene or lactones?

Answer: Thank you for your comment. We have modified the sentence to make it clearer [now L653]

Comment: L538. functional?

Answer: Thank you for your comment. We have corrected the typo [now L668]

Comment: L542. creating?

Answer: Thank you for your comment. We have corrected the typo [now L672]

Comment: L547. tended to higher. Simply “are higher” sounds better

Answer: Thank you for your comment. This sentence has been removed during rewording of the manuscript

Reviewer 2 Report

Title

The title is quite seductive but the statistical analyses does not support your Results and Discussion. Please consider a different title.

Introduction

L71-79: If you already know that P. undulatum has invasion mechanisms then, what is the novelty of your study? Please give a proper description of the new findings that you expect with your study. Why do you believe that by analyzing secondary metabolites you will gain new knowledge? Please provide a rationale.

L92: Please state your hypotheses here and not in the Discussion.

Material and Methods

L94: Please provide a better description of the level of degradation and/or fragmentation of your study area. With panel C of figure 1 I cannot imagine how degraded/fragmented is your area. The disturbance-invasibility hypothesis states that disturbance regime enhances the likelihood of natural communities to be invaded because disturbance provide new conditions and resources for alien species that are unsuitable or cannot be exploited by resident species.

L114 (Figure 1) and L152: Where can I see a description of the main characteristics of both the invaded and the remnant sites?

L120-123 and 128-131: I think both paragraphs are more appropriated for the Introduction section.

L133-142: I missed a description of your experimental site and not a general description of Eucalyptus. As this seems to be an 'observational study' with no interventions, please provide a description on the ecological status of your site i.e., degradation, fragmentation, level of inbreeding if possible. I mean, all factors that could have also restricted germination of Eucalyptus other than P. undulatum.

L143-148: This seems more appropriate for the Introduction or Discussion sections.

L153: I am not sure if you can use 'natural soil' as it. Did you sterilize your sample soils? Or you used it as it i.e., with pathogens, fungus, bacteria, other seeds, etc. Please clarify.

L154: I wonder how you can create this type of conditions by only using natural soil. What about light, moisture and/or temperature?

L160: Did you use pre-germinative treatments? or simply direct sowing?

L174: Were seeds collected from the tree or from the ground? From how many trees did you collect seed? Usually experiments conducted with seed from one or two trees had low germination capacity.

L180: Three or four substrates? Please correct.

L190: Why not spraying seeds with antifungal solution? Why not sterilize your substrates?

L194: What is this test? I am not familiar with it.

L303: Why not using repeated Measures here? You have daily data for germination.

L304: Why separated analyses? I guess you are interested into test the hypothesis of interaction between watering by substrate? Am I right? If so, you cannot separate analyses. This is the main reason of why I decided to reject your article. Please pay attention to the following comments.

L305: How did you analyze your data for germination? I guess in every dish plate you counted 1s and 0s for germinated and not germinated seeds. Then you obtained the percentage of germinated seeds per plate. If so, you must analyze percentage data with a generalized lineal model and not common ANOVA. Please specify your analyses.

L310: If you do not have a continuous variable such height and instead you have 'counts', you must analyze your data with general lineal model with a Poisson distribution and a logarithmic link function. If your raw data does not meet normality, this might imply that you have data with a different distribution. In this case, it could be a Poisson distribution. Please clarify what type of variable do you have here. I mean, TTC, TFC, and THC are continuous data, counts of 1s and 0s, percentages.

L318: Please specify if you used the arcsin transformation for percentage data or if you used a generalized lineal model for this type of data. You need to clearly describe your statistical analyses. I have identified several flaws here and this is the main reason of my decision to reject.

Results

L321: I wonder how can you affirm this sentence. As I mentioned before, you should have tested both factors but also the interaction between them.

L325-326: But you need letters in the Figure 2 (and also in Figure 3) to clearly see where are the differences.

L326-328: I wonder why you do not test the interaction between leachate and substrate. In your methods section you stated that your analyses were done 'separately' by substrate.  Why not to test the hypothesis of interaction between both factors? Later in the Discussion you consider this interaction and this is very confusing.

L330-332: What do you mean with the sentence 'did not significantly alter the GP'? What are you comparing here? Watering treatments within type of substrate? The theoretical GP for E. ovata reported in other studies? I am confused. You stated that analyses were done separately for substrate but you are combining the analyses here. Please clarify.

L333-334: I still do not understand how you made this comparison. In Table 1 you separate analysis for each substrate and lowercase letters are for watering treatment within each substrate. Then how you compare among substrates? Clarify.

L336 in parenthesis: This is confusing because I see in Table 1 that t50 has significant differences within watering treatments. Anyway, what calls my attention is that you do not have ‘a’ letters. You just have ‘ab’ and ‘b’ letters. This is rare. Please comment or clarify.

L339: OK, but I am now more confused. You have letters such as 'a', 'de', 'e', and 'ab' within treatments. I wonder where are the others letters? In the other substrates? If so, you have serious problems with your analyses. Where are the letters 'b' or 'c' for example?

L341: To support this sentence you should have analyzed the interaction between substrate and watering.

L381-383: I wonder how did you test this? Did you run orthogonal contrast analyses to compare subgroups within treatments? If so, please mention in the Methods. If not, you still have problems with your analyses. You cannot simply interpret your results from visual inspection, which I guess you did. Other issue. You stated that Dunnet test was used to test for differences among treatments. However, Dunnet test compare each of a number of treatments with a single control. In your case, which one is the control treatment?

L413-415: Similar to the other comment. Did you run orthogonal contrast analyses to support this contention? I guess no.

L433-434: Did you run individual ANOVAs for each variable?

L440: But calamenene is higher in I than in R. I see one asterisk indicating a p < 0,05. Clarify or correct.

L448: Please correct. It does not coincide with the Y-axis of Figure 8.

Discussion

L454-459: Why not declare these hypotheses at the end of the Introduction section? I have never been told that you had two hypotheses.

L466: But the second lowest percentage of germination was observed in seeds sowed in eucalyptus soil and watered with litter of P. undulatum (56%). I have doubts with your statistical analyses, but I see that germination of E. ovata was low when watering with this treatment. How can you explain this?

L495-496: But I see a clear difference when comparing the 90% of germination in the filter-paper by water treatments, with the 52% of germination in the Eucalyptus soil by water treatments. The problem here is the way you analyzed your data. 

L520-521: But I see in Figure 7 that 1,8 cineole was not different in the I than in the R sites. This is confusing.

L530-532: This is a repetition of results.

L543-544: Repetition of results.

L546-548: Repetition of results. Besides, what is the purpose of discussing non-significant results? What is the rationale here?

L549-554: What is the contribution of this paragraph? It seems just a description of 1,8 cineole and its main functions. But no differences were found between the I and R sites. Discuss this lack of difference.

L557-560: This is not Discussion. You are just repeating results but in other words.

L567-569: This is interesting but you do not properly describe both areas in terms of climatic variables, stand structure, level of degradation, fragmentation, etc. Is it possible that you add this type of information in the Methods section?

L570-573: This is an interesting discussion but I fell it is not well connected with your results. It is hard to follow the differences between mono- and sesquiterpernes. In Figure 7 monoterpenes are higher in the invaded area but in Figure 8 sesquiterpenes are higher in the remnant area (Eucalyptus site): Thus, which one are you referring here? Mono o sesquiterpenes? Clarify.

L574-581: I don't see much point in this paragraph. Why are you talking about drought and fire? How it is related with your study? Clarify.

Conclusions

L584: ‘may be due’. Avoid this expression. Conclusions must be a concise statement supported by your results.

L585: But secondary metabolites were higher in Eucalyptus extracts. How can you conclude this? Explain. You do not test abiotic stresses and cannot conclude this. Please consider changing this sentence.

L587: You are not testing any strategy here. Your work is about invasiveness of P. undulatum. Please do not lose your focus.

L589-590: This is just a comment and not a conclusion. Anyway, where can I find this 'broader understanding'?

References

You have an excessive number of references. Please leave the most relevant and if possible, the more updated. For example, do you need all seven references in Line 71 to talk about the invasive capacity of P. undulatum?

Author Response

We thank the reviewer for the useful comments. Here we provide a point-by-point response and a tracked-changes version of the manuscript (changes marked in red). 

Title

Comment: The title is quite seductive but the statistical analyses does not support your Results and Discussion. Please consider a different title.

Answer: Thank you for your comment and suggestion. We have modified the title as follows: Is the invasivenss of Pittosporum undulatum in Eucalypt forests explained by the wide ranging effects of its secondary metabolites?

 Introduction

Comment: L71-79: If you already know that Pundulatum has invasion mechanisms then, what is the novelty of your study? Please give a proper description of the new findings that you expect with your study. Why do you believe that by analyzing secondary metabolites you will gain new knowledge? Please provide a rationale.

Answer: Thank you for your comment. We have modified the paragraph to answer your comment [now L95-109]

Comment: L92: Please state your hypotheses here and not in the Discussion.

Answer: Thank you for your comment, we have modified the last part of the introduction and also the first part of the discussion. [now L110-118]

Material and Methods

Comment: L94: Please provide a better description of the level of degradation and/or fragmentation of your study area. With panel C of figure 1 I cannot imagine how degraded/fragmented is your area. The disturbance-invasibility hypothesis states that disturbance regime enhances the likelihood of natural communities to be invaded because disturbance provide new conditions and resources for alien species that are unsuitable or cannot be exploited by resident species.

Answer: Thank you, we have added some more information about the two areas [now L140-160] and we have also added a picture of each sampling site in Figure 1 (Panel D and E).

Comment: L114 (Figure 1) and L152: Where can I see a description of the main characteristics of both the invaded and the remnant sites?

Answer: Thank you, we have added some more information about the two areas (see answer to comment L94).

Comment: L120-123 and 128-131: I think both paragraphs are more appropriated for the Introduction section.

Answer: Thank you for your suggestion. We have moved and integrated these paragraphs in the Introduction section [now L60-76].

Comment: L133-142: I missed a description of your experimental site and not a general description of Eucalyptus. As this seems to be an 'observational study' with no interventions, please provide a description on the ecological status of your site i.e., degradation, fragmentation, level of inbreeding if possible. I mean, all factors that could have also restricted germination of Eucalyptus other than Pundulatum.

Answer: The dominant factor in all these sites is the presence of Pittosporum undulatum. The following sentence has been added to the text to clarify that other factors [now L154-160], including in breeding are highly unlikely. To make this clearer we have added the following:

“ Eucalyptus species are noteworthy in maintaining high genetic diversity even in fragmented woodland populations (Murray et al. 2017). Moreover, the trees in this study pre-date the fragmentation of the landscape and the disruption by invading P. undulatum. There has never been a record of a Eucalyptus seedling growing under a canopy of P. undulatum in the field in the forty years that this has been the subject of study (Gleadow and Ashton 1981).”

Comment: L143-148: This seems more appropriate for the Introduction or Discussion sections.

Answer: Thank you for your suggestion. We have moved these paragraphs in the Introduction section [now L86-94].

Comment: L153: I am not sure if you can use 'natural soil' as it. Did you sterilize your sample soils? Or you used it as it i.e., with pathogens, fungus, bacteria, other seeds, etc. Please

Answer: We have added a sentence in the text to clearly state this [now L174-175]. We have decided not to sterilise or alter the soil to observe the germination under nature-like conditions.

Comment: L160: Did you use pre-germinative treatments? or simply direct sowing?

Answer: We used direct sowing, without pre-germinative treatments

Comment: L174: Were seeds collected from the tree or from the ground? From how many trees did you collect seed? Usually experiments conducted with seed from one or two trees had low germination capacity.

Answer: Thank you for the comment, we have modified the sentence giving more information [now L195-196].

Comment: L180: Three or four substrates? Please correct.

Answer: Thank you for the comment. We have corrected the typo

Comment: L190: Why not spraying seeds with antifungal solution? Why not sterilize your substrates?

Answer: Thank you for pointing this out. We have removed this sentence as it was an error remaining from an earlier draft. Out of all petri dishes, only 3 seeds scattered in different treatments and substrates showed any sign of fungal infection. During our experimental design stage, we have decided to avoid spraying the seeds to keep conditions as natural as possible.

Comment: L194: What is this test? I am not familiar with it.

Answer: The squash test was conducted at the end of the experiment and involved crushing all the ungerminated seeds and observing the color and status of embryos. If the embryos were white, healthy and turgid, they were viable.

Comment: L303: Why not using repeated Measures here? You have daily data for germination.

Answer: Thank you for your comment. We did not use repeated measures but only at the end of the experiment, since taking the root length during the experiment could have damaged the seedlings sowed in soil.

Comment: L304: Why separated analyses? I guess you are interested into test the hypothesis of interaction between watering by substrate? Am I right? If so, you cannot separate analyses. This is the main reason of why I decided to reject your article. Please pay attention to the following comments.

Answer: Thank you for your comment. We have now changed the statistical analyses regarding the germination experiment [now L334-367].   

Comment: L305: How did you analyze your data for germination? I guess in every dish plate you counted 1s and 0s for germinated and not germinated seeds. Then you obtained the percentage of germinated seeds per plate. If so, you must analyze percentage data with a generalized lineal model and not common ANOVA. Please specify your analyses.

Answer: Thank you for your comment. We have corrected the statistical analyses for germination applying nonlinear mixed-effects (nlme) models for each species, defining total GP and t50 with the function SSLogis. The other indices (t0, tf and tf-t0) were analysed with the generalized linear model as suggested by you in the following comment (with family=Poisson, since the counts are discrete data with non-negative integer values). Thus, we have corrected and modified the paragraph of the germination analyses in the Materials and Methods and the relative section in the results [now L334-362].   

Comment: L310: If you do not have a continuous variable such height and instead you have 'counts', you must analyze your data with general lineal model with a Poisson distribution and a logarithmic link function. If your raw data does not meet normality, this might imply that you have data with a different distribution. In this case, it could be a Poisson distribution. Please clarify what type of variable do you have here. I mean, TTC, TFC, and THC are continuous data, counts of 1s and 0s, percentages.

Answer: Thank you for your comment. We have modified the text regarding how we obtained the ethanolic extract to make our methodology clearer. Here we have a continuous variable obtained with a quantification by HPLC of TTC, TFC and THC. Thus, we have carried out Shapiro and Levene tests and, since the assumption of normality was not met, we conducted a Kruskal-Wallis Test followed by a Dunnet post hoc test [now L368-372].  

Comment: L318: Please specify if you used the arcsin transformation for percentage data or if you used a generalized lineal model for this type of data. You need to clearly describe your statistical analyses. I have identified several flaws here and this is the main reason of my decision to reject.

Answer: Thank you for your comment, we have modified the text regarding the data. They are continuous variables and the relative amount of each identified terpene was calculated by determining each monoterpene and sesquiterpenes as percentage from the sum of all the identified terpene peak areas obtained by GC-MS, following criteria commonly used in literature [now L373-377].

 Results

Comment: L321: I wonder how can you affirm this sentence. As I mentioned before, you should have tested both factors but also the interaction between them.

Answer: Thank you for your comment. This section has been completely rewritten after running new statistical analyses.

Comment: L325-326: But you need letters in the Figure 2 (and also in Figure 3) to clearly see where are the differences.

Answer: Thank you for your comment, the Figures 2 and 3 have been modified according to the updated statistical analyses.

Comment: L326-328: I wonder why you do not test the interaction between leachate and substrate. In your methods section you stated that your analyses were done 'separately' by substrate.  Why not to test the hypothesis of interaction between both factors? Later in the Discussion you consider this interaction and this is very confusing.

Answer: Thank you for your comment. This section has been completely rewritten after running new statistical analyses.

Comment: L330-332: What do you mean with the sentence 'did not significantly alter the GP'? What are you comparing here? Watering treatments within type of substrate? The theoretical GP for Eovata reported in other studies? I am confused. You stated that analyses were done separately for substrate but you are combining the analyses here. Please clarify.

Answer: Thank you for your comment. This section has been completely rewritten after running new statistical analyses.

Comment: L333-334: I still do not understand how you made this comparison. In Table 1 you separate analysis for each substrate and lowercase letters are for watering treatment within each substrate. Then how you compare among substrates? Clarify.

Answer: Thank you for your comment. This section has been completely rewritten after running new statistical analyses (see answer to comment L305).

Comment: L336 in parenthesis: This is confusing because I see in Table 1 that t50 has significant differences within watering treatments. Anyway, what calls my attention is that you do not have ‘a’ letters. You just have ‘ab’ and ‘b’ letters. This is rare. Please comment or clarify.

Answer: Thank you for your comment. This has been changed after new statistical analyses

Comment: L339: OK, but I am now more confused. You have letters such as 'a', 'de', 'e', and 'ab' within treatments. I wonder where are the others letters? In the other substrates? If so, you have serious problems with your analyses. Where are the letters 'b' or 'c' for example?

Answer: Thank you for your comment. This has been modified after new statistical analyses

Comment: L341: To support this sentence you should have analyzed the interaction between substrate and watering.

Answer: Thank you for your comment. This has been changed after new statistical analyses

Comment: L381-383: I wonder how did you test this? Did you run orthogonal contrast analyses to compare subgroups within treatments? If so, please mention in the Methods. If not, you still have problems with your analyses. You cannot simply interpret your results from visual inspection, which I guess you did. Other issue. You stated that Dunnet test was used to test for differences among treatments. However, Dunnet test compare each of a number of treatments with a single control. In your case, which one is the control treatment?

Answer: See answer to comment L310

Comment: L413-415: Similar to the other comment. Did you run orthogonal contrast analyses to support this contention? I guess no.

Answer: See answer to comment L318

Comment: L433-434: Did you run individual ANOVAs for each variable?

Answer: Yes. We compared each identified compound between the two areas. the relative amount of each compounds is expressed as a percentage of total identified terpenes (TMTs + TSQTs) for both areas and then, for each compound for both areas, an ANOVA was run.

Comment: L440: But calamenene is higher in I than in R. I see one asterisk indicating a p < 0,05. Clarify or correct.

Answer: Thank you for your comment, we have corrected the sentence [now L558-559].   

Comment: L448: Please correct. It does not coincide with the Y-axis of Figure 8.

Answer: Thank you for your suggestion. We have modified the y-axis of figure 7 and 8

Discussion

Comment: L454-459: Why not declare these hypotheses at the end of the Introduction section? I have never been told that you had two hypotheses.

Answer: Thank you for your comment, we removed these lines from the Discussion section, and we have added them in the introduction to better explain our aims [now L110-119].   

Comment: L466: But the second lowest percentage of germination was observed in seeds sowed in eucalyptus soil and watered with litter of Pundulatum (56%). I have doubts with your statistical analyses, but I see that germination of Eovata was low when watering with this treatment. How can you explain this?

Answer: Thank you for your comment. This is no longer the case, as we have changed the statistical analyses, and for E. ovata seeds, the watering treatments and the relative combinations with substrates, were no longer considered since the exploration of marginal models showed no treatment significance.

Comment:  L495-496: But I see a clear difference when comparing the 90% of germination in the filter-paper by water treatments, with the 52% of germination in the Eucalyptus soil by water treatments. The problem here is the way you analyzed your data. 

Answer: Indeed we have now changed the analyses to address this issue.

Comment:  L520-521: But I see in Figure 7 that 1,8 cineole was not different in the I than in the R sites. This is confusing.

Answer: The amount of 1,8 cineole was higher in the R forest than in the I forest, but not statistically significant, as we have reported in the text [now L652].   

Comment: L530-532: This is a repetition of results.

Answer: Thank you for your comment, we have changed the sentence, removing the repetition of results.

Comment: L543-544: Repetition of results.

Answer: Thank you for your comment, we have modified the sentence, removing the repetition of results.

Comment: L546-548: Repetition of results. Besides, what is the purpose of discussing non-significant results? What is the rationale here?

Answer: Thank you for your comment, we have modified the sentence, removing the repetition of results. In addition, we decided to discuss also this non-significant result since the 1,8 cineole is one of the principal monoterpenes emitted by Eucalyptus spp., which has an important role against biotic attacks. This is in accordance with our results and second hypothesis

Comment: L549-554: What is the contribution of this paragraph? It seems just a description of 1,8 cineole and its main functions. But no differences were found between the I and R sites. Discuss this lack of difference.

Answer: See answer to comment L546-548

Comments: L557-560: This is not Discussion. You are just repeating results but in other words.

Answer: Thank you for your comment, we have modified the sentence, removing the repetition of results.

Comment: L567-569: This is interesting but you do not properly describe both areas in terms of climatic variables, stand structure, level of degradation, fragmentation, etc. Is it possible that you add this type of information in the Methods section? 695-698

Answer: Thank you for your comment. We have added some information relative to the two areas [now L693-717]. They are very close to each other and the only noticeable differences are in the presence/absence of P. undulatum plants and in the herbaceous layer of the soil. We have modified the text in the methods and discussion section to make it clearer.

Comment: L570-573: This is an interesting discussion but I fell it is not well connected with your results. It is hard to follow the differences between mono- and sesquiterpernes. In Figure 7 monoterpenes are higher in the invaded area but in Figure 8 sesquiterpenes are higher in the remnant area (Eucalyptus site): Thus, which one are you referring here? Mono o sesquiterpenes? Clarify.

Answer: Thank you for your comment. We have modified the text to make it clearer [now L717-721].   

Comment: L574-581: I don't see much point in this paragraph. Why are you talking about drought and fire? How it is related with your study? Clarify.

Answer: Thank you for pointing this out. We have removed this sentence as your suggestion.

Conclusions

Comment: L584: ‘may be due’. Avoid this expression. Conclusions must be a concise statement supported by your results.

Answer: Thank you for the comment, we have modified the text [now L734].

Comment: L585: But secondary metabolites were higher in Eucalyptus extracts. How can you conclude this? Explain. You do not test abiotic stresses and cannot conclude this. Please consider changing this sentence.

Answer: Thank you for the comment, we have changed the text [now L735].   

Comment: L587: You are not testing any strategy here. Your work is about invasiveness of Pundulatum. Please do not lose your focus.

Answer: Thank you for the comment, we have modified the text.

L589-590: This is just a comment and not a conclusion. Anyway, where can I find this 'broader understanding'?

Answer: Thank you for the comment, we have modified the text.

 References

You have an excessive number of references. Please leave the most relevant and if possible, the more updated. For example, do you need all seven references in Line 71 to talk about the invasive capacity of Pundulatum?

Answer: Thank you. We have modified and removed 40 references